# Research on a Method for Classifying Bolt Corrosion Based on an Acoustic Emission Sensor System

**DOI:** 10.3390/s24155047

**Published:** 2024-08-04

**Authors:** Shuyi Di, Yin Wu, Yanyi Liu

**Affiliations:** College of Information Science and Technology & Artificial Intelligence, Nanjing Forestry University, Nanjing 210037, China; dsy@njfu.edu.cn (S.D.); wuyin@njfu.edu.cn (Y.W.)

**Keywords:** corrosion level, wireless acoustic emission sensor, extreme learning machine, goose algorithm

## Abstract

High-strength bolts play a crucial role in ultra-high-pressure equipment such as bridges and railway tracks. Effective monitoring of bolt conditions is of paramount importance for common fault repair and accident prevention. This paper aims to detect and classify bolt corrosion levels accurately. We design and implement a bolt corrosion classification system based on a Wireless Acoustic Emission Sensor Network (WASN). Initially, WASN nodes collect high-speed acoustic emission (AE) signals from bolts. Then, the ReliefF feature selection algorithm is applied to identify the optimal feature combination. Subsequently, the Extreme Learning Machine (ELM) model is utilized for bolt corrosion classification. Additionally, to achieve high prediction accuracy, an improved goose algorithm (GOOSE) is employed to ensure the most suitable parameter combination for the ELM model. Experimental measurements were conducted on five classes of bolt corrosion levels: 0%, 25%, 50%, 75%, and 100%. The classification accuracy obtained using the proposed method was at least 98.04%. Compared to state-of-the-art classification diagnostic models, our approach exhibits superior AE signal recognition performance and stronger generalization ability to adapt to variations in working conditions.

## 1. Introduction

In recent years, with the advancement of new industrialization, higher requirements have been put forward for the high-quality development of infrastructure construction and significant technical equipment. High-strength bolts, as commonly used fasteners in pressure-bearing equipment such as bridges [1], steel rails [2], and ultra-high-pressure equipment, possess advantages such as good load-bearing performance, interchangeability, fatigue resistance, and resistance to loosening under dynamic loads. The repeated disassembly and various torque methods for installation impose incredibly high demands on high-strength bolts. The surface condition and thread accuracy of bolts directly affect the service life and safety of the equipment. However, with the increase in service life and the influence of external environments, the condition of bolts inevitably changes. For instance, corrosion is a significant factor affecting thread accuracy. Prolonged exposure to the marine atmospheric environment causes severe corrosion of bolts [3], resulting in difficulties in disassembly, insufficient pre-tensioning force between structural components [4], leading to loosening, detachment, or fatigue fracture of connected structures [5], posing new challenges for the normal service and routine maintenance of structural components, and even triggering major safety accidents. Therefore, the development of bolt corrosion detection and classification technology is of great significance.

Scholars at home and abroad have proposed many technical means for detecting the connection status of bolts, such as traditional sound detection, strain gauge detection, and piezoelectric sensor technology. However, these methods need to improve their accuracy and have defects, such as the need for regular calibration. In recent years, with the development of detection technology, Gao et al. [6] have used digital torque wrenches for bolt group loosening detection, which can display torque values, automatically save data, and are easy to operate. Daniel et al. [3] calculated the fiber Bragg grating (FBG) wavelength drift caused by bolt loosening based on the deformation of FBG curvature sensors caused by bolt loosening. They ultimately monitored bolt loosening using FBG curvature sensors. Deng et al. [7] proposed a flange bolt ultrasonic measurement model, and Yuan et al. [8] effectively measured the pre-tensioning force of bolts based on a method whereby the ultrasonic transducer and bolt can be separated. Lao et al. [9] comprehensively considered the influence of camera focal length, shooting angle, and lighting conditions on bolt detection and measurement. They proposed a method for measuring bolt loosening rotation angles that can be used under different imaging conditions. However, the above methods are similar to conventional structural health monitoring systems. For example, although fiber Bragg grating sensors have the advantages of stability and reliability, they require a large number of transmission lines, leading to a high initial investment. Once the wires are damaged, the system will stop working, and the maintenance and repair time and economic costs are high. Digital image processing technology cannot accurately detect slight bolt loosening or make accurate judgments on minor changes in corrosion levels.

With the development of wireless communication technology and electronic technology, structural health monitoring systems based on WASNs [10] have attracted widespread attention. AE technology, firstly, can detect dynamic defects such as material fracture and crack propagation without relying on external energy and can evaluate the safety of objects without entering their interiors. Secondly, it can operate at a long distance, monitor the expansion of equipment defects, and obtain real-time information about structural defects as they vary with load, time, temperature, and other factors. It has more advantages in detecting the status of bolts.

Therefore, we intend to study a bolt corrosion classification system based on a WASN to more accurately detect and assess corrosion. This method mainly collects stress waves generated inside the bolt due to external vibrations and propagates them to the receiving probe along the flange. Artificial intelligence technology is then used to process the collected data for corrosion level classification. The recent literature has also mentioned the use of AE equipment to detect devices and materials such as wood and steel bars. Almeida et al. [11] associated AE signals with the damage mechanism sources of the composite materials under study using machine learning methods, obtaining necessary information such as the location and intensity of each damage mechanism. Zhang et al. [12] selected five features that were most sensitive to the uncertainty and complexity of AE signals generated during wood fracture and studied their classification performance using the Empirical Mode Decomposition-Discrete Wavelet Transform-Linear Discriminant Analysis classification model, ultimately achieving high accuracy in feature extraction and classification results and realizing intelligent detection of internal wood damage. Seye et al. [13] used AE techniques to evaluate the fracture process of corroded and healthy reinforced concrete (RC) beams, revealing intuitively the influence of corrosion damage on the bending failure process of RC beams. However, research on the effects of bolt corrosion is still limited, and even if it is involved, it only focuses on the changes in bolt pre-tensioning caused by corrosion. For example, Tian [14] conducted in-depth research on the AE characteristics before the fracture of high-strength bolts, aiming to monitor the changes in the health status of high-strength bolts due to corrosion in real-time, but did not specifically improve the diagnosis and classification of bolt corrosion levels. Therefore, it is not easy to make accurate judgments and take maintenance measures earlier according to the situation.

In summary, this paper proposes a bolt corrosion classification system based on a WASN. Low-power and high-precision WASN nodes are used to receive AE signals, and different feature parameters are collected for in-depth research. The main highlights of the results can be summarized as follows:(1)The gateway uses the ReliefF feature selection algorithm to screen the optimal features, thereby improving the accuracy of identification;(2)The ELM (Extreme Learning Machine) model is used for corrosion level diagnosis and classification, and the GOOSE algorithm is used to optimize parameters;(3)Experimental results show that the classification model based on AE sensors designed for bolt corrosion levels of 0%, 25%, 50%, 75%, and 100% outperforms traditional methods with higher identification accuracy.

## 2. WASN Bolt State Non-Destructive Testing System

### 2.1. Trial Specimen Design and Sensor Selection

In the experiment, five high-strength bolts with a diameter of 54 mm of 10.9 grade conforming to national standard NB/T31082-2016 [15] were selected. Five bolts were subjected to electrochemical accelerated corrosion treatment separately. [16]; all the specimens were immersed in NaCl solution, and the positive electrode of the external DC power supply was connected to one end of the bolt. Meanwhile, another conductive cathode material was placed in the solution and connected to the negative electrode of the DC power supply. A positive constant direct current is applied between the bolt and the cathode material to accelerate corrosion.

Under the action of the external electric field, the electrochemical oxidation reaction of the half-cell, as shown in Equation (1), occurs on the bolt surface. The half-cell electrochemical reduction reaction, as shown in Equation (2), occurs on the surface of the anode material.
(1)$${\mathrm{Fe}-2e}^{-}\to {\mathrm{Fe}}^{2+}$$
(2)$$O_{2}+2H_{2}O+4e^{-}\to 4{\mathrm{OH}}^{-}$$

The relationship between the theoretical mass loss of the bolt *m_t_* (g) and the energizing time *t* (s) is shown below.
(3)$$m_{t}=\frac{M_{\mathrm{Fe}}I}{2F}•t$$
where *F* is the Faraday constant (96,485 C/mol), *I* is the corrosion current intensity (A), and $M_{\mathrm{Fe}}$ is the relative atomic mass of *Fe* (55.847). Corrosion current density is generally 0.045~3 mA/cm^2^. The current density of the experiment is 2 mA/cm^2^, and the calculated current should be maintained at about 0.22 A.

The degree of corrosion of bolts is divided and prepared according to the different times of energizing. The bolt corrosion degree is 100% when the power is on for 30 h, and the bolt corrosion degrees are 0%, 25%, 50%, and 75% when the power is on for 0, 6, 12, 18, and 24 h, respectively. By using the definition of rust grade in the national standard for verification, that is, the coverage area and adhesion state of the oxide on the specimen surface are observed by the visual method. We found that the ratios of rust area to the total surface area of the bolt are consistent with the classification results obtained through the time-differentiated power-on experiment.

To show this more vividly, Figure 1 illustrates the microscopic states of bolts with different degrees of corrosion under a 20× magnifying glass.

Because the AE signal frequency of normally functioning bolts is between 100 kHz and 125 kHz, and the signal frequency caused by corrosion-induced deformation falls within the range of 125 kHz to 250 kHz, for this experiment, the UT1000 type wideband AE sensor was selected. It exhibits an ideal response within the range of 1 MHz.

To conduct experiments, the prepared bolt specimens are sequentially screwed into a flange with an inner diameter of 5.2 cm, outer diameter of 24 cm, and thickness of 4 cm.

Regular working bolts emit AE signals ranging from 0 kHz to 125 kHz. Signals indicating deformation due to corrosion occur in the frequency range of 125 kHz to 150 kHz [17]. Therefore, the selected AE sensor is required to have excellent measurement characteristics within this frequency range, along with high sensitivity and response speed. Hence, for this experiment, the UT1000 wideband AE sensor was chosen. It exhibits highly ideal response characteristics up to 1 MHz, ensuring that it can capture signals even if their frequencies vary within the range of 0 kHz to 150 kHz. This makes the UT1000 suitable for broader applications.

### 2.2. WASN Hardware System Framework Design and Installation Layout Scheme

Firstly, the system is powered by a battery composed of six series-connected nickel-hydrogen batteries rated at 1.2 V each. The UT1000-type AE sensors collect bolt AE data from two independent channels, both of which are amplified using OPA627 operational amplifiers. The amplified signals are then converted from analog to digital using the ultra-low power ADC AD7356 and output to the STM32F405RG chip for signal reading. After data preprocessing and storage, a low-power, long-distance, and cost-effective wireless communication network is built using LoRa modules based on SX1278. Figure 2 illustrates the composition structure of the AE node.

It is noteworthy that in the power supply modules converting 5 V to 3.3 V and 2.5 V, the ADM7170 voltage regulator chip is chosen to maintain low noise levels in the output voltage. The OPA627 chip is employed to amplify the original AE signals, enhancing the circuit’s noise suppression capability and improving the signal-to-noise ratio of the AE signals. The frequency-shift keying modulation method is utilized in the wireless communication module, which also exhibits good noise immunity. Inspired by X. Lang et al. [18], who proposed a multiscale convolutional neural network based on kurtosis and Kullback–Leibler divergence to significantly improve noise immunity, this paper adopts a threshold voltage method to filter noise. AE signals exceeding the threshold voltage are identified as valid signals, while signals below the threshold are considered noise.

When the bolt and flange specimen is subjected to dynamic excitation or collision friction, elastic waves are generated and propagate through the structure. After reflection and refraction, these waves reach the surface of the connected specimen and are detected and received by the AE sensors. In the experiment, the sensors need to be in close contact with the surface of the object. To minimize the signal loss caused by sensor selection and placement, the sensors are installed on the upper and lower nuts of the bolt using magnetic suction cups. The two probes are aligned with the flange plate as the boundary, positioned on the same straight line. They are noticeably distributed at the far and near ends of the flange plate surface. The center distance from the flange surface to the near-end probe is 1.5 cm, and to the far-end probe is 5.5 cm. This setup enables differential operation on the AE signals received by the two probes, thereby minimizing interference from other factors, fully leveraging the advantages of the AE sensors, reducing system measurement errors, and collecting purer signals. The connection method of the specimens is illustrated using bolts with corrosion levels of 100% and 0% as examples, as shown in Figure 3.

Coupling agents are added at the junction between the sensor and the specimen surface to increase the tightness of the connection. At the beginning of the experiment, referring to the acoustic source simulation method of the stress wave tester for wood, use a handheld rubber mallet to strike the upper nut, and use the interaction between the rubber mallet and the upper nut as the sound source. See Figure 4.

As the stress waves generated by vibration propagate through the specimen structure, they are collected by the AE sensor after reflection and refraction at the specimen surface. Following the above steps, experiments are conducted on five types of bolt samples. The AE signal acquisition process for bolt samples with corrosion levels of 25% and 50% is illustrated in Figure 5 below.

### 2.3. AE Data Acquisition and Analysis

All AE data collected by the WASN will be sent to the gateway and presented in the upper computer interface. The following diagram illustrates the AE data display interface, which depicts the AE waveform. 

In addition, a large number of acoustic emission characteristic parameters can be extracted from the acoustic emission signals, and some of these characteristic values are described as follows: (1)Threshold: Set according to the mean, variance, and other statistical parameters of the signal and the distribution of noise, with repeated adjustments to find the optimal threshold;(2)Amplitude: Maximum voltage threshold in decibels (dB), used for wave source type identification;(3)Rise time: The time interval between the acoustic emission signal first exceeding the threshold voltage and reaching the maximum voltage amplitude, used for noise identification;(4)Duration: Time difference between the first and last occurrences of the acoustic emission signal exceeding the threshold voltage;(5)Ringing count: Number of times the acoustic emission signal exceeds the threshold voltage;(6)Power: Area under the energy envelope spectrum or the sum of squared sample values, also used for identifying the type of wave source.

The characteristic parameters will be automatically summarized into a table for easy compilation of the dataset for bolt condition assessment. The analysis will be demonstrated using the AE waveforms of bolts with corrosion levels of 25% and 50% as examples. See Figure 6.

The sampling period is 1 s. A total of 10,000 samples are taken in this second. Therefore, the horizontal coordinate range is 0 to 10,000. If the duration of the effective acoustic emission signal is short, as shown in the figure above, the horizontal coordinate of the acoustic emission waveform cannot reach 10,000. Upon comparison, it is observed that bolt samples with higher levels of corrosion exhibit denser data, allowing for the presentation of more AE characteristic points.

## 3. Initial Framework of Bolt Corrosion Diagnosis Method 

Using the AE signal characteristic parameters collected by the WASN as the sample set for five groups of bolts with different degrees of corrosion, Figure 7 presents the basic conceptual framework for diagnosing and classifying corrosion levels.

The main logic involves first performing a differential operation on the AE signals received by the two probes to minimize interference from other factors, reduce system measurement errors, and ensure signal purity. Then, using the ReliefF algorithm for feature selection, there are 12 features that the AE signal can express, but not all of them have positive and decisive significance for determining the corrosion level of the bolt. Therefore, a feature selection algorithm is needed. Initially, all features are ranked based on their weights, and starting with only the feature with the highest weight, additional features are gradually added in order of weight to the most basic ELM (Extreme Learning Machine) model. The accuracy is calculated each time, and the selected features with the highest accuracy are considered the most relevant AE signal features for the corrosion level. These selected features are then used to train the model to adequately represent the characteristics of the original data. Subsequently, a classification diagnosis model is established using the ELM. Since the initial weights and thresholds of the ELM are randomly generated, the GOOSE algorithm is introduced to optimize its parameters and train the best diagnostic model, thereby improving generalization ability and achieving the highest possible accuracy. The superiority of the GOOSE-ELM algorithm is validated by comparing it with various other algorithms.

## 4. Bolt Corrosion Level Diagnosis and Classification Method Design

### 4.1. Data Preprocessing

Firstly, ensure that five high-strength bolts with a grade of 10.9, complying with the Chinese standard NB/T31082-2016, have been sufficiently corroded to the corresponding levels of 0%, 25%, 50%, 75%, and 100%, and can be clearly distinguished by appearance. The bolts with a diameter of 54 mm are successively tightened into the inner aperture of a flange plate with a diameter of 52 mm, an outer diameter of 24 cm, and a thickness of 4 cm for experimentation. AE parameters are collected through the WASN nodes. Vaseline facilitates the connection between the node and the flange plate to maximize the transmission of valid information in the signal. Each sample is sampled 100 times, and for each AE signal, including near-end energy/amplitude/duration/ring count, far-end energy/amplitude/duration/ring count, and the difference in energy/amplitude/duration/ring count between near and far ends, a total of 12 features are analyzed subsequently. 

Each feature contains five × 100 data points. Scatter plots of the near- and far-end data of the amplitude and duration features are shown in Figure 8 and Figure 9.

### 4.2. Feature Selection Based on ReliefF

Feature selection is a crucial task in the data preprocessing stage, aiming to identify the most discriminative and informative feature variables from raw data to enhance the performance of classification models. The ReliefF algorithm, a feature weighting technique [19], is derived from the Relief algorithm initially designed to address binary classification problems [20]. It evaluates the importance of features by computing their similarity and dissimilarity and assigns a weight to each feature, indicating its contribution to the classification task. Based on these weights, features are sorted and selected, making ReliefF widely applicable in addressing regression problems with continuous target attributes [21].

Firstly, the raw data require basic preprocessing, encompassing data cleansing, handling missing values, and standardization, ensuring data quality and consistency, which are essential prerequisites for feature selection. Following data verification, the AE dataset can be normalized to the [0, 1] range using the formula: (4)$${x^{*}}_{i}=\frac{x_{i}-x_{\min}}{x_{\max}-x_{\min}}\begin{array}{cc} & i=1,2,\ldots \end{array}$$

Let *x_i_* denote the original data, with *x_max_* and *x_min_* representing the maximum and minimum values, respectively, among the input feature data. According to the formula, the normalized result *x^*^_i_* can be calculated.

Next, assuming the training dataset is denoted by *D*, consisting of samples belonging to *Y* categories, and randomly selecting sample *R_i_* belonging to class *C*, with a sampling frequency of m and a threshold of *δ* for feature weights, the output is the feature weights, *T*, with all feature weights initially set to zero.

Subsequently, from the training set *D*, a sample *R* is randomly selected, and *k* nearest neighbor samples *Hj* (*j* = 1, 2,…, *k*) belonging to the same class as *R*, termed “correct nearest neighbors”, are identified. Additionally, *k* nearest neighbor samples *Mj(C)* from each sample not belonging to the same class as *R*, termed “incorrect nearest neighbors,” are identified. This process is repeated *m* times.

Finally, all features required for weight determination are input into the following formula: (5)$$W(A)=W(A)-\frac{\sum_{j=1}^{k}\mathrm{diff}(A,R,H_{j})}{mk}+\sum_{C\notin class(R)}\frac{\left[\frac{p(C)}{1-p(Class(R))}\sum_{j=1}^{k}diff(A,R,M_{j}(C))\right]}{mk}$$

Update the feature weights according to the formula.

In the above equation, *p(C)* represents the proportion of this category, and *p(Class(R))* denotes the proportion of the category of the randomly selected sample. $diff(A,R_{1},R_{2})$ represents the difference between sample *R1* and sample *R2* on feature *A*, and *M_j_(C)* represents the *jth* nearest neighbor sample in class *C*, with further elaboration provided in Equation (3).
(6)$$diff(A,R_{1},R_{2})=\left\{ \begin{array}{l} \frac{\left|R_{1}[A]-R_{2}[A\right]}{\max(A)-\min(A)}\begin{array}{cccc} if & A & is & continuous \end{array}\\ \begin{array}{ccc} & & \;0\begin{array}{cccc} & & & \begin{array}{cccc} \;\;if & A & is & discreate\&\&R_{1}[A]=R_{2}[A] \end{array} \end{array} \end{array}\\ \begin{array}{ccc} & & \;1\begin{array}{cccc} & & & \begin{array}{cccc} \;\;if & A & is & discreate\&\&R_{1}[A]=R_{2}[A] \end{array} \end{array} \end{array} \end{array}\right.$$

The algorithm operates efficiently, without imposing restrictions on data types or assuming specific distributions, making it versatile for handling large-scale datasets and diverse classification tasks.

Therefore, in this study, the ReliefF algorithm is selected to compute the weights of each feature. The results obtained when integrated into the ELM model reflect features with relatively low correlation, that is, features with weights below a certain threshold will be removed.

Considering the randomness in the selection of sample R during the algorithm’s execution, different random selections will yield different weight results. Therefore, an averaging approach is employed. The main program is executed four times, and the obtained five sets of weight data exhibit a similar trend, as illustrated in Figure 10.

The 12 features on the x-axis are arranged as follows: near-end energy/amplitude/duration/ring count, far-end energy/amplitude/duration/ring count, and the difference in energy/amplitude/duration/ring count between near end and far end. Summarizing and averaging the results reveals that the difference in energy between the near and far ends is the primary factor reflecting the characteristics of AE signals under bolt corrosion, directly indicating variations in signal intensity at different levels. Following this, the amplitude difference between the near and far ends ranks second, with the weights of the third and fourth features being relatively close in magnitude. This suggests that both the amplitude and energy at the near end are crucial determinants of corrosion severity. Overall, the ranking of the superiority of all AE features is shown in Table 1. 

In practical operations, the optimal number of feature sets is incremented from one, and each resulting combination is sequentially input into the default parameter original ELM classification model to obtain approximate classification accuracy results. Eventually, the analysis reveals that when using the top seven ranked feature combinations, the classification accuracy reaches a maximum of 93.4%. Subsequently, the accuracy begins to decrease. Therefore, all subsequent operations in this paper will be based on the selected seven features for classification decisions. See Figure 11.

The processing of the dataset using the ReliefF algorithm enables the determination of the importance level of feature attributes, which is crucial for classifying rust levels. This provides valuable insights that can reduce misdiagnosis errors and enhance the speed and accuracy of diagnostics.

### 4.3. GOOSE-ELM Classification Algorithm

#### 4.3.1. Original Extreme Learning Machine (ELM)

The neural network is a computational model that mimics the human brain’s neural system, consisting of multiple neurons that communicate through connections to transmit information [22]. Traditional neural networks require significant computational resources and time during training and are prone to becoming stuck in local optima. Therefore, a new neural network algorithm called Extreme Learning Machine (ELM) [23] has emerged. ELM is a fast and simple neural network model commonly used for classification and regression tasks. Compared to traditional neural networks, ELM has advantages such as fast training speed, ease of implementation, and strong generalization ability.

Initially, the weights from the input layer to the hidden layer are initialized. The general structure of a single-hidden-layer feedforward neural network consists of the input layer, a hidden layer, and an output layer, where there is a complete connection between neurons from the input layer to the hidden layer and from the hidden layer to the output layer.

Suppose there are *n* neurons in the input layer corresponding to *n* input variables, each mapped to one neuron in the hidden layer; there are *m* neurons in the output layer corresponding to *m* output variables. Without loss of generality, let the connection weights *w_ij_* between the input layer and the hidden layer be randomly initialized, with each weight selected randomly from a uniform distribution.
(7)$$w=\left[\begin{array}{l} w_{11}\begin{array}{cc} & w_{12}\begin{array}{cc} \ldots & w_{1n} \end{array} \end{array}\\ w_{21}\begin{array}{cc} & w_{22}\begin{array}{cc} \ldots & w_{2n} \end{array} \end{array}\\ \begin{array}{cccc} & & \;\;\ldots & \ldots \end{array}\\ w_{l1}\begin{array}{cc} & w_{l2}\begin{array}{cc} \ldots & w_{ln} \end{array} \end{array} \end{array}\right]$$

The term *w_ij_* represents the connection weight between the *ith* neuron in the input layer and the *jth* neuron in the hidden layer.

Next, the activation function for the hidden layer is defined. The activation function for hidden layer neurons is typically a nonlinear function, such as the sigmoid function, ReLU function, etc.

The next step involves computing the output of the hidden layer: the input samples are multiplied by the weight matrix *W* and passed through the activation function to obtain the output of the hidden layer, typically denoted as *H*. Let *g(·)* represent the activation function for the hidden layer neurons. Then, the output of the hidden layer *H* is calculated as:(8)H=g(XW+b)

Here, *X* represents the feature matrix of input samples, with a size of *N × n*, where *N* is the number of samples and *n* is the dimension of features.
(9)$$X=\left[\begin{array}{l} x_{11}\begin{array}{cc} & x_{12}\begin{array}{cc} \ldots & x_{1N} \end{array} \end{array}\\ x_{21}\begin{array}{cc} & x_{22}\begin{array}{cc} \ldots & x_{2N} \end{array} \end{array}\\ \begin{array}{cccc} & & \;\ldots & \ldots \end{array}\\ x_{n1}\begin{array}{cc} & x_{n2}\begin{array}{cc} \ldots & x_{nN} \end{array} \end{array} \end{array}\right]$$

*W* is the weight matrix from the input layer to the hidden layer, with a size of *n ×*
l, where l is the number of hidden layer neurons; *b* is the bias vector of the hidden layer, with a size of 1 × l.
(10)$$b=\left[\begin{array}{l} b_{1}\\ b_{2}\\ \ldots \\ b_{l} \end{array}\right]$$

Then, for the calculation of the output layer weights, when *g(·)* is infinitely differentiable, the connection weights *β* between the hidden layer and the output layer can be obtained by solving the least squares solution, that is
(11)$$\beta =H^{+}Y$$

Given that *H+* is the Moore–Penrose pseudoinverse of *H*, and *Y* is the label of the samples, with a size of l
*× m*.

The neural network’s output *T* can be obtained from the structural diagram:(12)$$T={\left[t_{1},\ldots ,t_{Q}\right]}_{m*Q}$$
(13)$$t_{j}={\left[t_{1j},\ldots ,t_{mj}\right]}^{T}={\left[\begin{array}{l} \sum_{i=1}^{t}\beta_{i1}g(w_{i}x_{j}+b_{i})\\ \sum_{i=1}^{t}\beta_{i2}g(w_{i}x_{j}+b_{i})\\ \begin{array}{cccc} & & \ldots & \end{array}\\ \sum_{i=1}^{t}\beta_{im}g(w_{i}x_{j}+b_{i}) \end{array}\right]}_{m*l},(j=1,2,\ldots ,Q)$$

In which, $w_{i}=[w_{i1},w_{i2},\ldots ,w_{in}],x_{j}={\left[x_{1j,}x_{2j},\ldots ,x_{nj}\right]}^{T}$.

Combining the above equations, we can obtain *T′*:(14)$$T^{\prime }=H\beta$$
where *T′* is the transpose matrix of *T*, and *H* is the matrix representing the output of the neural network’s hidden layer, which can be expanded as follows:(15)$$H(w_{1},\ldots ,w_{i},b_{1},\ldots ,b_{l},x_{1},\ldots ,x_{Q})={\left[\begin{array}{l} g(w_{1}*x_{1}+b_{1})\begin{array}{cc} & g(w_{2}*x_{1}+b_{2}) \end{array}\begin{array}{cc} \ldots & g(w_{l}*x_{1}+b_{l}) \end{array}\\ g(w_{1}*x_{2}+b_{1})\begin{array}{cc} & g(w_{2}*x_{2}+b_{2})\begin{array}{cc} \ldots & g(w_{l}*x_{2}+b_{l}) \end{array} \end{array}\\ \begin{array}{cccc} & \;\;\ldots & & \end{array}\\ g(w_{1}*x_{Q}+b_{1})\begin{array}{cc} & g(w_{2}*x_{Q}+b_{2}) \end{array}\begin{array}{cc} \ldots & g(w_{l}*x_{Q}+b_{l}) \end{array} \end{array}\right]}_{Q*l}$$

The final step involves the practical application of the model: given new input samples, the output result *T* of the ELM can be directly obtained through the output of the hidden layer, *H*, and the weight matrix *β* of the output layer.

Key features of the ELM algorithm include the random initialization of hidden layer weights and the direct calculation of output layer weights. Only one computation of the pseudoinverse of the weight matrix is required, eliminating the need for iteration or backpropagation. This dramatically accelerates training speed, making it highly efficient for handling large-scale data. Additionally, ELM mitigates overfitting issues by initializing weight matrices randomly, enabling rapid data fitting while maintaining good generalization during testing.

Finally, ELM implementation is relatively straightforward, requiring minimal parameter tuning and complex computations. This makes ELM an efficient machine-learning algorithm.

#### 4.3.2. Enhanced ELM with GOOSE Algorithm

The GOOSE algorithm is a novel metaheuristic intelligent optimization algorithm that simulates the different behaviors of a flock of geese in resting and alert states to find the optimal solution. First proposed by Hamad and Rashid [24], this algorithm adapts the resolution and search speed of the search space to rapidly and accurately find the optimal solution, continuously repeating this process until a predetermined number of iterations is reached or a stopping criterion is met. It features fast convergence and high solution accuracy, making it an excellent optimization algorithm. In the improved GOOSE-ELM algorithm, the GOOSE algorithm is used to adjust the weights and bias terms of the ELM to improve classification performance.

The inspiration for the GOOSE algorithm comes from the behavior of geese during rest and alertness. When resting, geese gather in flocks, with one goose assigned as a guard. To avoid falling asleep and to remain vigilant, the guard goose lifts one leg and balances on it while carrying a small stone. If the guard goose falls asleep, the stone falls, waking it up. At the same time, other geese will loudly honk if they notice any noise or activity, alerting the flock to potential threats.

Initially, the algorithm initializes by constructing the *X* matrix, representing the positions of search agents analogous to geese positions. In each iteration, the fitness of each search agent is determined using a normalized benchmark function, and the fitness of the other agents is compared to obtain the best fitness and position. The fitness of each current row is compared, and during the iteration, the fitness before this row is returned.

Next, a variable *rnd* is determined using conditions and random variables, with its value uniformly distributed according to the number of iterations to balance between the exploration and exploitation phases. Given a specified random variable, assume *rnd* = 0.5, indicating a 50% probability of locating the development or exploration iteratively using conditions that are evenly distributed between exploration and development. Additionally, *pro* and *coe* variables are introduced to identify effective equations and determine whether the discrimination value is less than or equal to 0.17.

During development, the stone weight carried by geese, ranging from 5 g to 25 g, is randomly generated using the following formula in each iteration:(16)$$W_{it}=randi([5,25],1,1)$$

Let *T_G_* denote the time required for the stone to fall to the ground. It undergoes random variations between dimensions in each iteration of the loop:(17)$$T_{G}=rand(1,\dim)$$

The time taken for this sound to reach any individual goose in the group is denoted as *T_E_*:(18)$$T_{E}=rand(1,\dim)$$

Then, the average time taken for the sound to propagate from the source to reach an individual goose during the entire iteration process is determined:(19)$$T_{\mathrm{ave}}=\frac{\sum T}{2\cdot \dim}$$

Considering the constraints on parameters, the value of *pro* must be greater than 0.2, and *W_it_* should be greater than or equal to 12. *F* represents the falling speed of the object, and 9.81 represents the acceleration due to gravity. Therefore, to ensure the protection and awakening of the goose group, the following formula should be used: (20)$$F=\frac{T_{G}\sqrt{W_{it}}}{9.81}$$

The distance of sound propagation is calculated by multiplying the speed of sound in the air *v* by the propagation time *T_E_*. Next, we calculate the distance between the guarding goose and another goose that is either resting or feeding. It is essential to multiply by 1/2 because we only need the time for sound to propagate outward without considering the time for the sound to return.
(21)$$D=\frac{1}{2}VT_{E}$$

We set an optimal *X_it_*, where its value equals the falling object’s speed, *F*, plus the square of the time multiplied by the distance between the two geese, *D*.
(22)$$X_{\left(it+1\right)}=F+D\cdot {T_{\mathrm{ave}}}^{2}$$

If the values of the parameters *pro* and *W_it_* do not meet the above conditions, then a new *X_it_* is searched for. In this case, *F* will be redefined as:(23)$$F=\frac{T_{G}W_{it}}{9.81}$$

For the exploration phase, if one of the geese wakes up immediately and emits a sound to protect the other individuals, then it enters the exploration phase. That is, when *rnd* is less than 0.5, the following equation is applied:(24)$$X_{\left(it+1\right)}=\alpha M_{T}\cdot rand(1,\dim)+P$$
where dim is the number of dimensions in the problem, *P* is the best position found within the search area (*X_it_*). The variable *α* takes values from 0 to 2 and decreases significantly with the number of iterations in the loop:(25)$$\alpha =2-(\frac{2\cdot loop}{M_{IT}})$$

In the algorithm, the use of formulas allows for the continual discovery of new positions for the goose flock within the search space, depending on the values of different parameters. Through ongoing updates and iterations, the optimal parameters can ultimately be identified for output, establishing an ELM model based on the GOOSE algorithm. Figure 12 illustrates the algorithmic process of GOOSE-ELM, detailing how the GOOSE algorithm optimizes the weights and biases of ELM. Combined with the model construction approach outlined earlier, this presents a comprehensive diagnostic classification mechanism.

GOOSE-ELM offers several advantages over traditional ELM. Firstly, by optimizing weights and biases, GOOSE-ELM can better adapt to the characteristics of the dataset, thereby improving classification performance. Secondly, GOOSE-ELM exhibits high robustness and is capable of handling data containing noise or outliers. Additionally, it boasts faster training speeds and lower computational complexity, making it an ideal choice for processing large-scale datasets.

## 5. Results and Discussion

### 5.1. Results

Summarize the above process and carry out the operations. Firstly, the sample data collected by the sensor are preprocessed, normalizing them into dimensionless data. Next, utilize the ReliefF algorithm to select the most representative seven features based on the processed data. Finally, train the optimal classification diagnostic mechanism through the GOOSE-ELM optimization and classification algorithm, ultimately achieving high-accuracy classification of input random samples. 

Splitting 500 sample data in a 3:7 ratio into a test set and a training set, we take the test set, which consists of 150 samples. Each type of rust accounts for 30 samples, totaling five types of rust. Calculate the confusion matrix to evaluate the classification performance of the model [25].

Figure 13 and Figure 14 depict, respectively, the confusion matrix and a comparison between the actual values and the algorithm’s classification results.

From the graph, it can be observed that the classification accuracy (the ratio of the correctly predicted samples to the total samples) reaches 98.04%. One instance of class two rust was misclassified as class one, one example of class three rust was misclassified as class five, and one instance of class four rust was misclassified as class three. Overall, there is little difference between the actual values and the final classification results.

Further analysis of other evaluation metrics for classification models includes:

*TP* (True Positive): Samples predicted as accurate and valid. *FP* (False Positive): Samples predicted as accurate but which are actually false. *TN* (True Negative): Samples predicted as false and which are actually false. *FN* (False Negative): Samples predicted as false but which are actually valid.

The accuracy mentioned above refers to the ratio of the number of correctly predicted samples to the total number of samples, and the formula is as follows:(26)$$A=\frac{TP+TN}{TP+TN+FP+FN}$$

Precision is the proportion of actual positive samples among all samples predicted as accurate. It is a suitable method for evaluating the quality of optimistic class predictions. The calculation formula is:(27)$$P=\frac{TP}{TP+FP}$$

The precision of the five rust classes is 100%, 96.7%, 96.7%, 96.7%, and 100%, respectively. Therefore, the average precision is 98.02%.

Recall is the proportion of actual positive samples among all accurate positive samples in the dataset. It helps focus on reducing false negatives. The calculation formula is:(28)$$R=\frac{TP}{TP+FN}$$

The precision of the five rust classes is 96.8%, 100%, 96.7%, 100%, and 96.8%, respectively. The average recall rate is calculated to be 98.06%.

F1 score is the harmonic mean of precision and recall, which comprehensively considers the performance of both. In situations where precision and recall are equally important, the F1 score is a valuable metric. It is particularly suitable for imbalanced datasets. The calculation formula is:(29)$$F1=\frac{2PR}{P+R}$$

The F1 score of this algorithm is 0.9804.

For the solution of the ROC curve [26], considering it is a multi-class problem, the one-vs.-rest strategy is utilized to transform the multi-class problem into multiple binary classification problems. Then, for each binary classification problem, the ROC curve and AUC are calculated, and they are combined to compute the average ROC curve and AUC. AUC represents the area under the ROC curve and is commonly used to measure the performance of classifiers across all classes. The AUC value ranges from 0 to 1, where a value closer to 1 indicates better classifier performance, while a value closer to 0.5 indicates poorer performance. The shape of the ROC curve helps understand the classifier’s performance at different thresholds. The closer the curve is to the top-left corner, the better the classifier’s performance. See Figure 15.

The plot indicates that the classification results for all five levels of rust severity are approaching 1 indefinitely, suggesting that GOOSE-ELM performs well across all categories.

### 5.2. Verification

To verify the effectiveness of the ReliefF feature selection algorithm and the superiority of the GOOSE-ELM classification model, 500 sets of sample data were randomly divided into training and testing sets in a 7:3 ratio. The primary verification approach was as follows:(1)Using the Pearson correlation coefficient method and the ReliefF feature selection method, the importance of features was ranked from the aspects of correlation and weight, respectively. The number of features was incrementally increased and incorporated into the original ELM classification model, where the highest accuracy represented the optimal feature dimension for each method. The comparison between the maximum accuracy and kappa coefficient at the optimal feature dimension demonstrates the merits of the two approaches;(2)With the feature selection algorithm determined to use the ReliefF algorithm, four different classification algorithm models were separately applied for analysis. The superiority of the various models was judged based on accuracy, precision, recall, and F1 score.

#### 5.2.1. Feature Selection Performance Analysis

The Pearson correlation coefficient [27] is an essential tool for measuring the strength of the relationship between two variables. Its value is obtained by dividing the covariance between variables by the product of their standard deviations. A value closer to 1 or −1 indicates a stronger linear relationship between the variables, while a value closer to 0 indicates a weaker linear relationship between them.
(30)$$r=\frac{C\mathrm{ov}(X,Y)}{\sqrt{D(X)}*\sqrt{D(Y)}}$$
where *X* and *Y* represent two separate feature parameters, denoting the standard deviation, and *Cov(X, Y)* represents the corresponding covariance. For the twelve features mentioned in the text, calculating the correlation between each pair yields the heatmap shown below (Figure 16): 

It is noticeable that there is a strong correlation between features such as the near/far-end energy difference and the near/far-end amplitude difference, the near-end energy and the near/far-end duration difference. Therefore, based on the magnitude of correlation with other features as the criterion for ranking features, the features are sorted. Starting from the feature with the highest correlation, each subsequent feature is added one by one and incorporated into the ELM classification model. The accuracy and kappa coefficient of this method are calculated and compared with the ReliefF algorithm, as shown in Table 2:

Among them, accuracy refers to the proportion of correctly classified samples to the total number of test samples. The kappa coefficient is a metric used for consistency testing and measuring classification accuracy, with values ranging from −1 to 1. It indicates the proportion by which the algorithm’s classification results reduce classification errors compared to completely random classification results. The higher the value, the better the sorting effect. It not only considers the accuracy of the model’s predictions but also compares them with random guessing, making it highly interpretable. Through comparison, it can be observed that the ReliefF algorithm’s accuracy and kappa coefficient are both higher than those of the Pearson correlation coefficient method, proving its superiority.

#### 5.2.2. GOOSE-ELM Algorithm Performance Analysis

Based on Figure 7 and Figure 8, the distribution of data points for different rust categories on the scatter plot can be roughly considered clustered, with noticeable gaps between clusters. Therefore, we consider using K-means clustering and hierarchical clustering methods to classify the data and compare the classification results with those of GOOSE-ELM. The classification results are shown in Table 3.

It can be observed that the accuracy of both clustering algorithms is not high. This is because, in practical computations, the dataset exhibits a complex structure with considerable overlap between clusters, and the gaps between different categories are not sufficiently large. Apart from the features displayed in the scatter plot, the spatial distribution differences of various rust categories are not distinct enough. Therefore, clustering algorithms are not the optimal choice.

Given that the ReliefF algorithm is used for feature ranking, the same sample dataset is input into four different algorithm models: the traditional ELM classification algorithm, Hunter–Prey Optimization classification algorithm [28] (HPO-ELM), Sparrow Swarm Algorithm classification algorithm [29] (SSA-ELM), and Goose Optimization classification algorithm (GOOSE-ELM). 

Carnivorous animal hunting strategies inspire the HPO-ELM algorithm. It stimulates the chase-and-capture process of prey to find the optimal combination in the hyperparameter space, maximizing the accuracy or other performance metrics of Extreme Learning Machine (ELM) models. It exhibits good generalization ability but may suffer from local optima. The SSA-ELM algorithm optimizes parameters in the ELM algorithm by simulating the behavior of sparrows. This enhances the algorithm’s search capabilities in the solution space and its ability to discover global optima, but it has some dependence on parameter selection.

The evaluation metrics for each classification model, including accuracy, precision, recall, and F1 score, are computed and summarized in Table 4.

To better illustrate the differences in classification performance among the four algorithm models, a three-dimensional bar chart corresponding to the four evaluation indicators is presented as follows in Figure 17:

Compared to the other three algorithms, the GOOSE-ELM algorithm exhibits significant advantages across all four indicators. It boasts high accuracy, indicating the model’s overall solid predictive capability. Additionally, it demonstrates high precision, indicating an accurate prediction of positive samples. The high recall rate signifies the model’s ability to correctly predict the number of positive samples relative to all actual positives, reflecting predictive solid power for positive instances. With a high F1 score, serving as a comprehensive metric, the GOOSE-ELM classification algorithm emerges as a model with robust performance across all aspects.

When comparing the SSA-ELM Sparrow Swarm Classification Algorithm and the GOOSE-ELM Goose Optimization Algorithm individually, the former ranks second after GOOSE-ELM. The evaluation uses the CEC 2005 standard test functions, which are widely used to assess the performance of optimization algorithms. This evaluation employs convergence curves as evaluation metrics and includes a total of 25 test functions, each with varying levels of difficulty and characteristics, covering a wide range of optimization challenges. See Figure 18. 

The F1 function is a simple, unimodal, continuous test function with its global optimum at x_i_ = 0, as depicted in the figure. The shape of the test function and the location of the optimum can be intuitively observed from the graph. Since F1 is a simple quadratic function, its plot typically resembles a unimodal curve. See Figure 19. 

The F5 function is a multimodal test function. From the graph, it can be observed that the F5 function is a non-convex function with many local minima. The global optimum is typically located at the highest peak of the function plot. In contrast, local optima are the local minima points on the function plot, as shown in the figure, where multiple local minima exist. See Figure 20. 

The F8 function is one of the most challenging test functions for intelligent algorithms. Due to its multiple peaks, intelligent algorithms often struggle to find the global optimum and easily become trapped in local minima during the optimization process. Therefore, this function poses a significant challenge for intelligent optimization algorithms. Its theoretical minimum value is −12,569.5. Conventional intelligent optimization algorithms can achieve optimization values around −7000, such as the SSA optimization algorithm. However, the proposed GOOSE optimization algorithm in this paper achieves an optimization value lower than −8000 for the F8 function, which is closer to the theoretical value. Compared to the SSA algorithm, the GOOSE optimization algorithm demonstrates faster convergence speed and can obtain relatively accurate results with fewer iterations. Therefore, the GOOSE algorithm is also more efficient. See Figure 21. 

The F21 function is a mixed-composite function composed of multiple sub-functions, making its analysis quite complex. However, as observed from the graphs, both algorithms exhibit suitable adaptability mechanisms capable of adjusting search strategies and parameter settings based on the characteristics of the search space and the shape of the function, thereby quickly finding the optimal solution. Nevertheless, the GOOSE algorithm still maintains a lower optimization value. Therefore, under this test function, the GOOSE algorithm remains superior.

## 6. Conclusions

This study investigated a bolt corrosion classification system based on a Wireless Acoustic Sensor Network (WASN). Utilizing the ReliefF algorithm, the optimal features were selected from the 12 AE signal parameters of the bolts. The system optimized the SVM operational parameters using the GOOSE optimization algorithm to construct the GOOSE-ELM classification model, achieving high accuracy in classification performance. The following conclusions can be drawn from the comparison and evaluation of the classification results:(1)The ReliefF algorithm demonstrated high efficiency in optimizing the selection process of multiple mixed features. In practical operations, by inputting the top seven ranked features into the recognition model, SVM achieved an accuracy rate of over 98.04% for the collected data;(2)The GOOSE-ELM model, optimized by the GOOSE algorithm for ELM classification model parameters, exhibited the best performance. The classification diagnosis system based on this algorithm showed good steady-state accuracy, recall rate, and F1 score for classifying the degree of bolt corrosion.

Through multiple experiments and practical applications, it can be fully demonstrated that this system can effectively classify the degree of bolt corrosion accurately. It outperforms the most advanced classification methods currently available in terms of classification diagnostic performance and generalization ability, adapting well to fluctuations in working conditions.

In the future, breakthroughs can be made in a comprehensive analysis of multiple parameters, integrating with other sensors for holistic assessment of bolt health, thus avoiding potential limitations of single sensors. Overall, precise monitoring of rust conditions can prevent accidents caused by bolt failures, enhancing safety and reliability. Predictive maintenance can reduce equipment downtime, further improving the availability and operational efficiency of production equipment. The application of WASN not only aids in bolt rust detection but also provides a new technological pathway for industrial automation and smart manufacturing, fostering the development of intelligent and automated production.

## Figures and Tables

**Figure 1 sensors-24-05047-f001:**
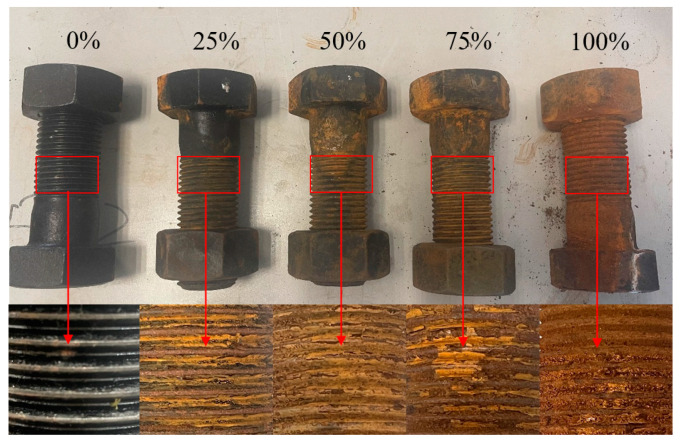
Illustration of the five types of bolt samples.

**Figure 2 sensors-24-05047-f002:**
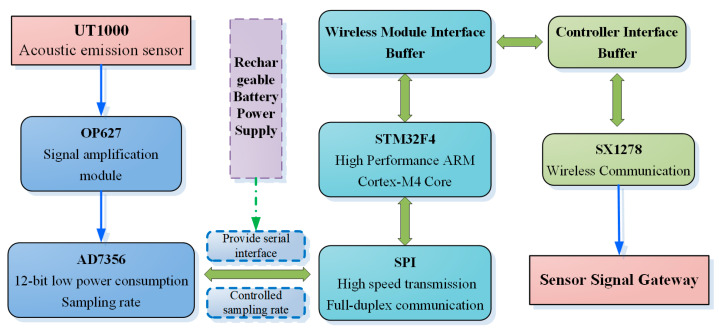
The framework of wireless AE node configuration.

**Figure 3 sensors-24-05047-f003:**
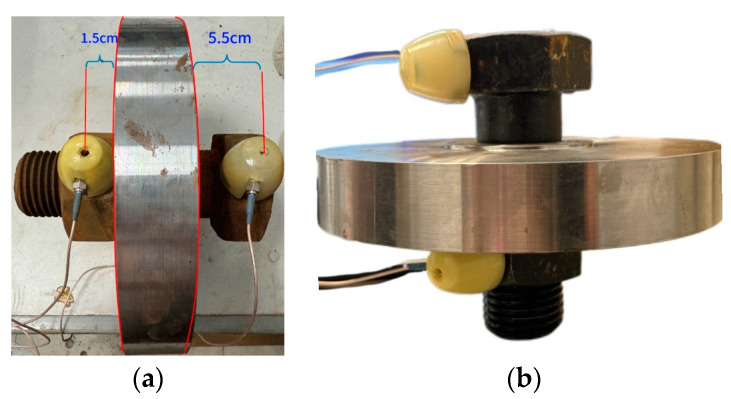
Schematic diagram of the bolt and specimen connection method: (**a**) Connection method for bolt specimens with a corrosion grade of 100%; (**b**) Connection method for bolt specimens with a corrosion grade of 0%.

**Figure 4 sensors-24-05047-f004:**
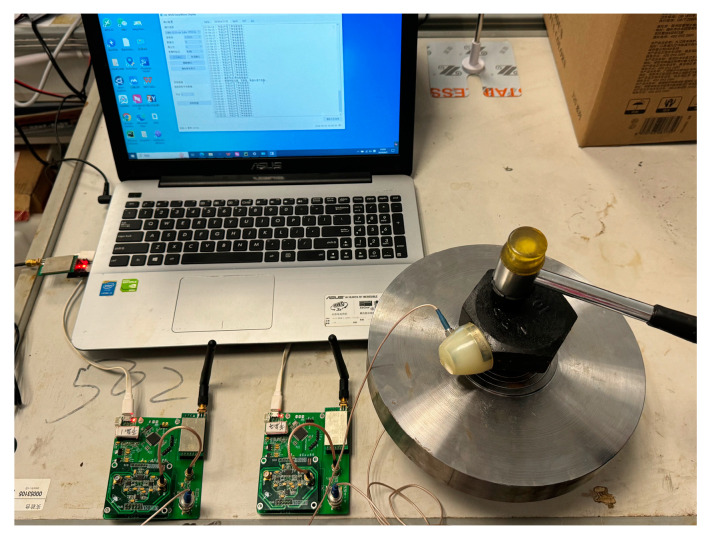
Diagram of the external excitation process.

**Figure 5 sensors-24-05047-f005:**
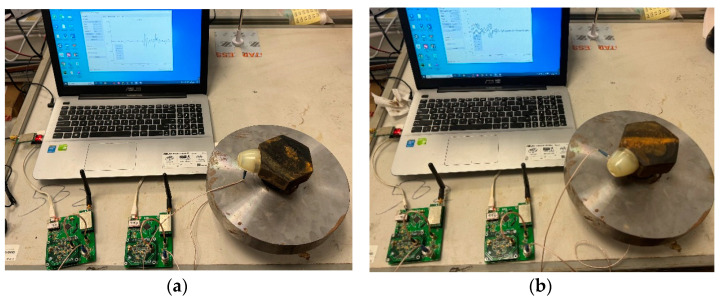
Diagram of AE signal acquisition process: (**a**) The AE signal acquisition process for bolt samples with corrosion levels of 25%; (**b**) The AE signal acquisition process for bolt samples with corrosion levels of 50%.

**Figure 6 sensors-24-05047-f006:**
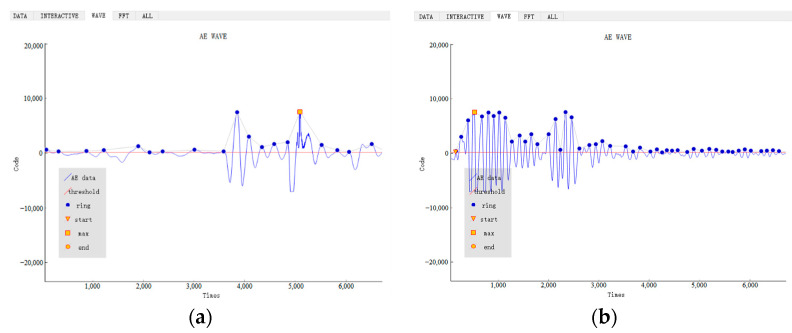
Schematic diagram of AE waveforms: (**a**) AE waveforms of bolts with corrosion levels of 25%; (**b**) AE waveforms of bolts with corrosion levels of 50%.

**Figure 7 sensors-24-05047-f007:**
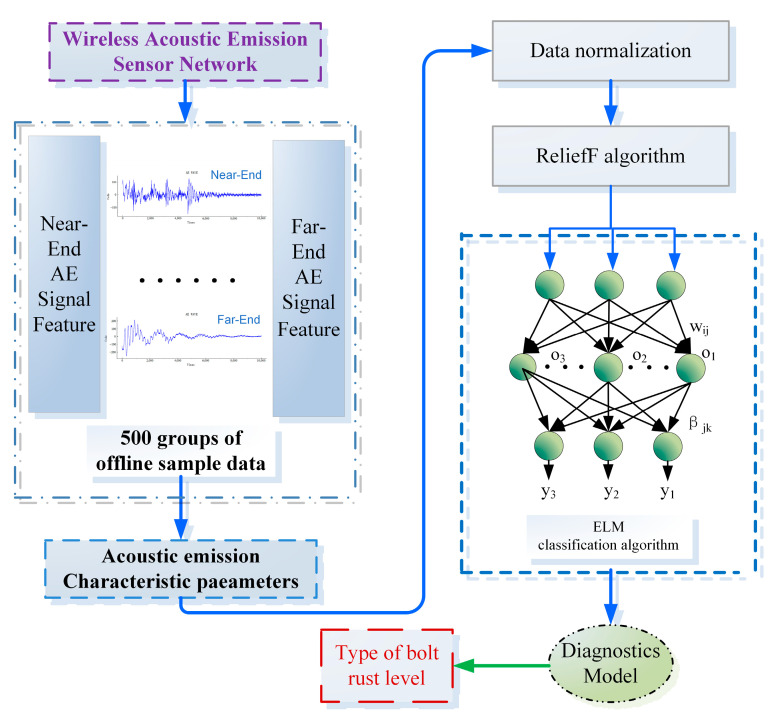
Basic conceptual framework of the classification system.

**Figure 8 sensors-24-05047-f008:**
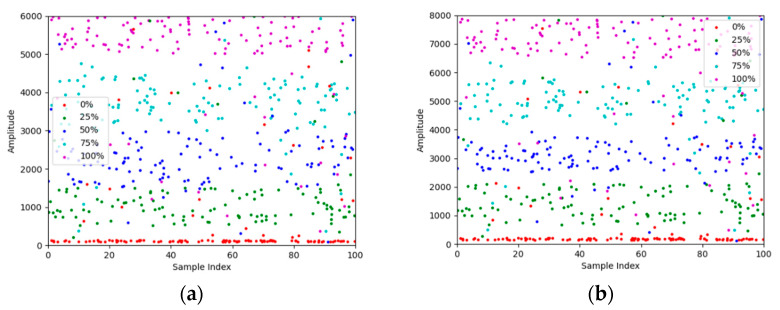
Scatter plot of amplitude data. (**a**) Scatter plot of amplitude near-end data; (**b**) Scatter plot of amplitude far-end data.

**Figure 9 sensors-24-05047-f009:**
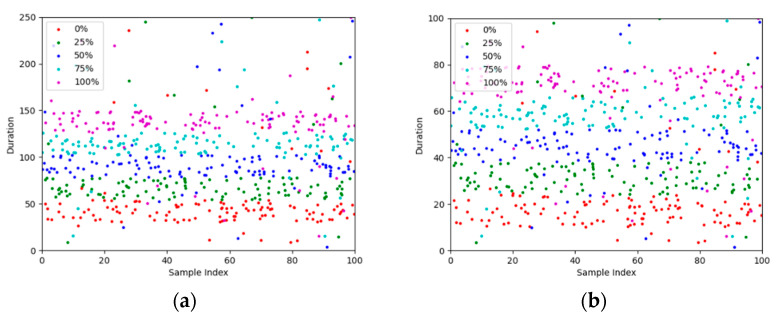
Scatter plot of duration data: (**a**) Scatter plot of duration near-end data; (**b**) Scatter plot of duration far-end data.

**Figure 10 sensors-24-05047-f010:**
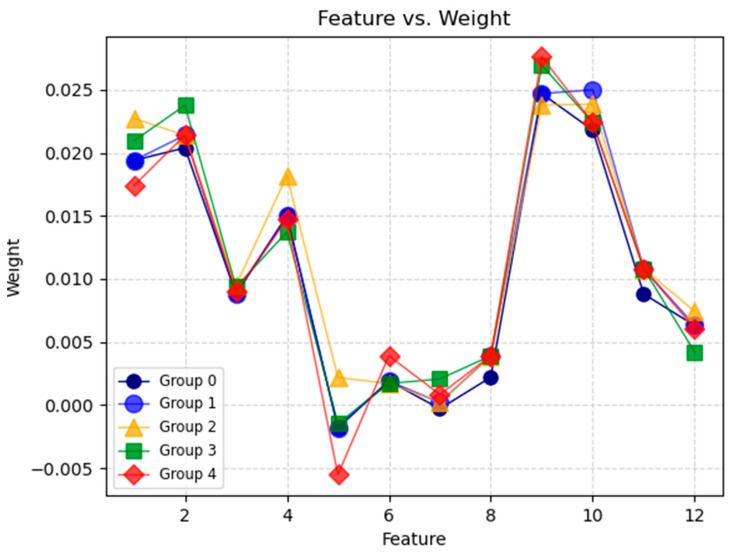
Illustrates the weights of the 12 features.

**Figure 11 sensors-24-05047-f011:**
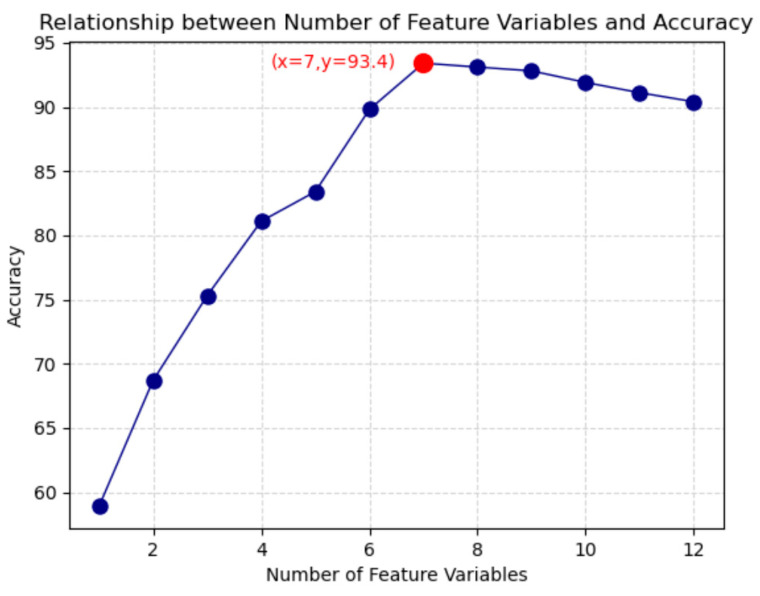
Relationship between the number of features and accuracy.

**Figure 12 sensors-24-05047-f012:**
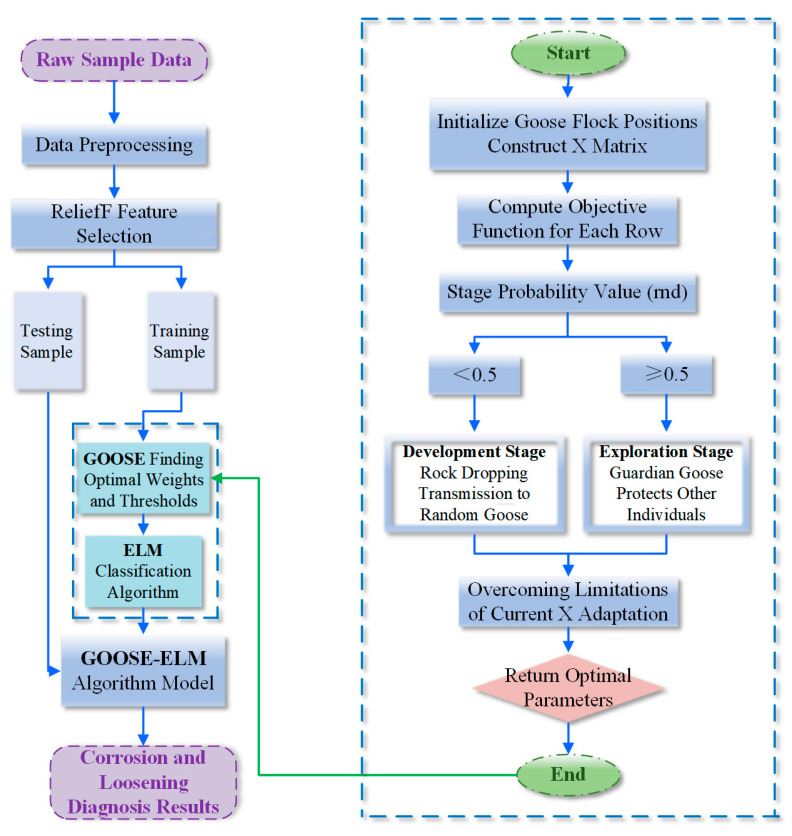
GOOSE-ELM algorithm flowchart.

**Figure 13 sensors-24-05047-f013:**
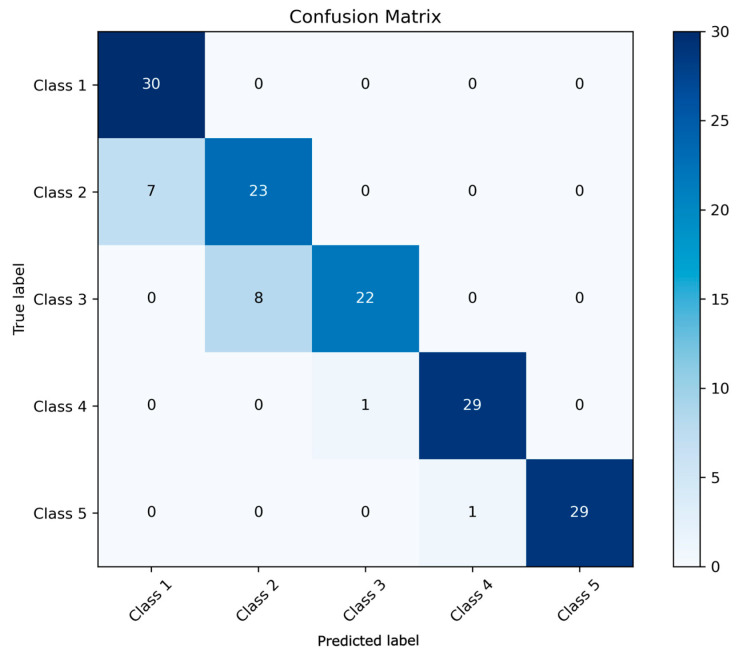
Confusion matrix of GOOSE-ELM algorithm classification results.

**Figure 14 sensors-24-05047-f014:**
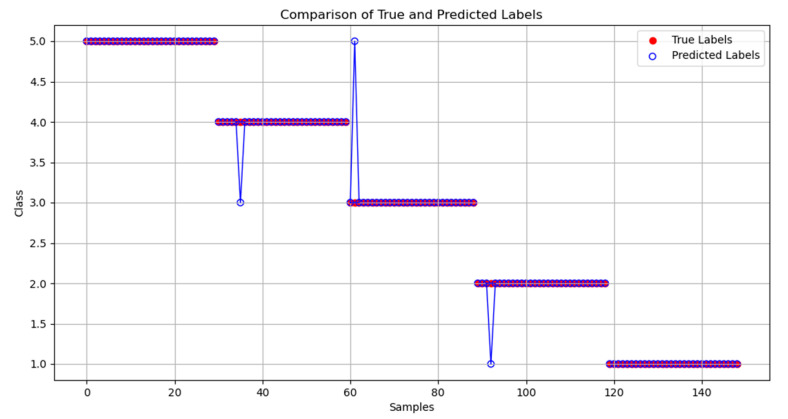
Comparison between classification results and actual classification.

**Figure 15 sensors-24-05047-f015:**
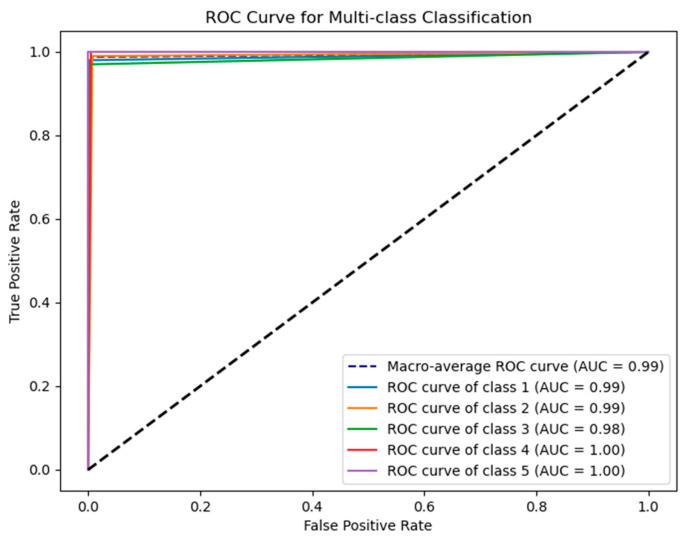
GOOSE-ELM algorithm ROC curve.

**Figure 16 sensors-24-05047-f016:**
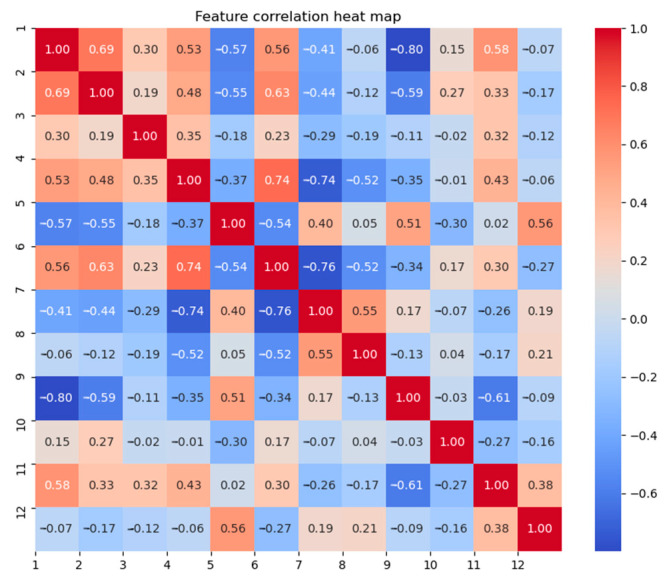
Heatmap of the 12 features.

**Figure 17 sensors-24-05047-f017:**
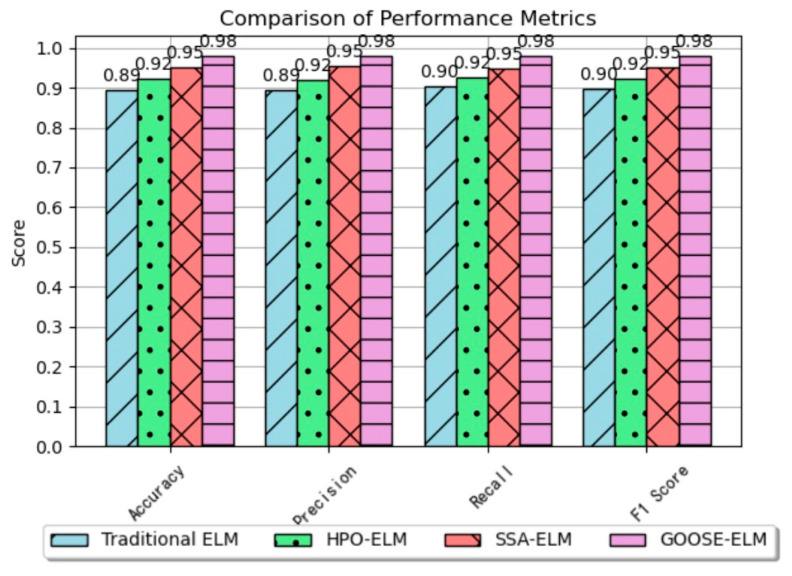
Comparison of the four evaluation indicators.

**Figure 18 sensors-24-05047-f018:**
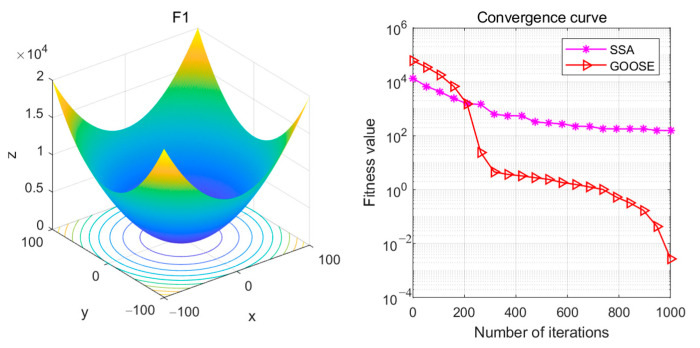
F1 test functions and convergence curves.

**Figure 19 sensors-24-05047-f019:**
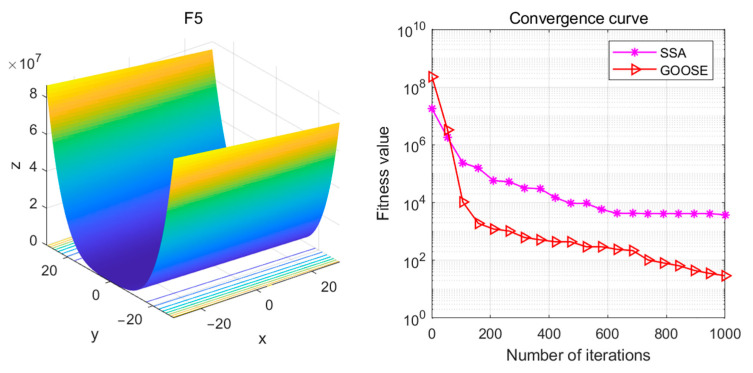
F5 test functions and convergence curves.

**Figure 20 sensors-24-05047-f020:**
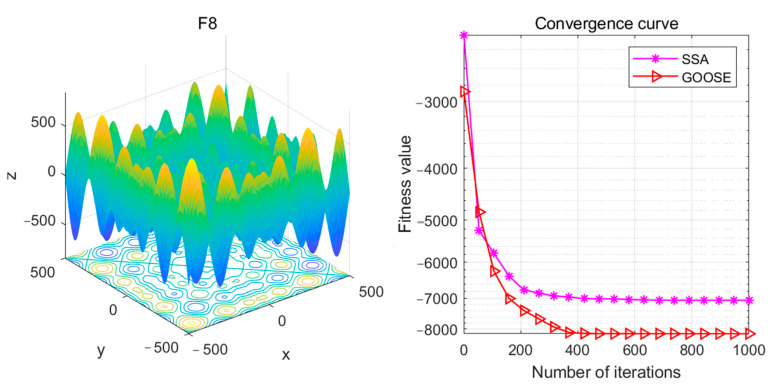
F8 test functions and convergence curves.

**Figure 21 sensors-24-05047-f021:**
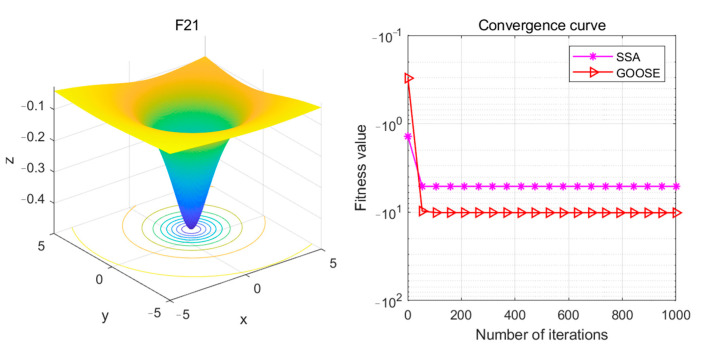
F21 Test functions and convergence curves.

**Table 1 sensors-24-05047-t001:** Ranking of AE feature weights.

Feature Selection Algorithm	Weighted Ranking of AE Features
ReliefF	(1) Near/far-end energy difference; (2) Near/far-end amplitude difference; (3) Near-end amplitude; (4) Near-end energy; (5) Near-end ringing count; (6) Near/far-end duration difference; (7) Near-end duration; (8) Near/far-end ringing count difference; (9) Far-end ringing count, (10) Far-end amplitude; (11) Far-end duration; (12) Far-end energy

**Table 2 sensors-24-05047-t002:** Comparison of results between two feature selection methods.

Feature Selection Methods	Dimension Selected	Accuracy	Kappa
Pearson	9	89.11	0.857
ReliefF	7	98.04	0.975

**Table 3 sensors-24-05047-t003:** Comparison of results between two clustering algorithms and the GOOSE-ELM algorithm.

Classification Method	K-Means Clustering	Hierarchical Clustering	GOOSE-ELM
Accuracy	64.67	67.33	98.04

**Table 4 sensors-24-05047-t004:** Results of four classification algorithms on four evaluation metrics.

Classification Algorithms	Accuracy (%)	Precision (%)	Recall (%)	F1 Score
Traditional ELM	89.33	89.34	90.20	0.8977
HPO-ELM	92.22	92.00	92.44	0.9222
SSA-ELM	95.10	95.33	94.87	0.9510
GOOSE-ELM	98.04	98.02	98.06	0.9804

## Data Availability

The datasets presented in this article are not readily available because the data are part of an ongoing study.

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
