# Peer review of "Research on a Method for Classifying Bolt Corrosion Based on an Acoustic Emission Sensor System"

_sensors, 2024, doi:10.3390/s24155047_

Round 1
Reviewer 1 Report
Comments and Suggestions for Authors
The manuscript by S. Di et al. is based on an investigation using AE monitoring and it-based data to classify the corrosion of bolts by focusing the corrosion levels. The topic and content match the journal sensors well and the findings have values. I would like to recommend it for publication. However, several points must be classified or improved before publication.
1) If you provide an abbreviation for a term, like AE for acoustic emission, then just use it after the place where the abbreviation was given; On the contrary, if you did not give the full spell of some term, please don’t use the abbreviation, like EMD-DWT-LDA.
2) Experimental: some important information was missing.
a) Line 110-112: how to quantitatively prepare the samples with different corrosion levels of 0, 25, 50, 75, and 100%?
b) Line 117-120: how does the author know or evidence the specific sources with the corresponding frequency range of 100-125, 125-250 kHz? Assuming this is true, then why would the authors choose a wideband sensor to detect a process with such a narrow range of frequency characteristics?
c) How to filter out the noise during AE monitoring experiments? Threshold-based, or frequency filtering-based? More details should be provided.
d) Line 167-171: how were the AE parameters automatically extracted in the AE acquisition system used and how to define a waveform?
e) In Fig. 5: how to know that the AE waveforms demonstrated here were not from the noises?
3) Results and discussion: How long for the testing time for each AE monitoring on different samples? What do you think that the AE data shown in this study were generated by the corrosion evolution or corrosion-induced deformation?
4) Some microscopy analysis is greatly expected to show the initial microstructure, as well as to evidence the corrosion behavior under different corrosion levels.
Author Response
Reviewer #1
Firstly, we fully appreciate the careful review and valuable suggestions from you. We have made the following modifications in response to your suggestions:
Comments 1 : [ If you provide an abbreviation for a term, like AE for acoustic emission, then just use it after the place where the abbreviation was given; On the contrary, if you did not give the full spell of some term, please don’t use the abbreviation, like EMD-DWT-LDA.]
Response 1: First of all, thank you for pointing this out. We agree with this comment. Everywhere except in abstracts and references, we abbreviated acoustic emission to AE, and the Wireless Acoustic Emission Sensor Network was all shortened to WASN. The substitutions have been marked in red in the revised manuscript. For phrases in which the full spell was never mentioned and explained in the article, the full name was retained after modification. For example, "EMD-DWT-LDA", we replaced it with "Empirical Mode Decomposition-Discrete Wavelet Transform-Linear Discriminant Analysis", You can find them on page 2, lines 49, 83, etc.
Comments 2 : [Experimental: some important information was missing. a) Line 110-112: how to quantitatively prepare the samples with different corrosion levels of 0, 25, 50, 75, and 100%? b) Line 117-120: how does the author know or evidence the specific sources with the corresponding frequency range of 100-125, 125-250 kHz? Assuming this is true, then why would the authors choose a wide-band sensor to detect a process with such a narrow range of frequency characteristics? c) How to filter out the noise during AE monitoring experiments? Threshold-based, or frequency filtering-based? More details should be provided. d) Line 167-171: how were the AE parameters automatically extracted in the AE acquisition system used and how to define a waveform? e) In Fig. 5: how do know that the AE waveforms demonstrated here were not from the noises?]
Response 2: Thanks so much for your comment.
a) In the experiment, the definition of corrosion levels is established by controlling the electrification time of bolts to categorize corrosion levels based on the resulting quality degradation of the bolts. The bolt corrosion degree is 100% when the power is on for 30 hours, and the bolt corrosion degree is 0%, 25%, 50% and 75% when the power is on for 0, 6, 12, 18 and 24 hours, respectively. Simultaneously, upon conclusion of the experiment, the classification results obtained are verified against the visual observation method described in the national standard, which involves assessing the coverage area and adherence status of oxidation layers on the specimen surfaces. It was found that the ratio of corroded area to the total surface area of the bolt also corresponds to the percentage defined for corrosion levels.
On page number 3, lines 109-134, a specific method for quantitative preparation of samples of different rust grades is given.
b) In-depth literature review was conducted to further refine the range of data referenced in the article. Finally, according to reference 15 on page 4, line 147, the frequency range of the signal emitted by the bolt during normal operation can be more accurate to 0khz ~ 125khz, and the frequency range of the deformation signal caused by corrosion is 125khz ~ 150khz. Based on this, the UT1000 possesses a wide bandwidth, capable of covering signals within the range of 0 kHz to 150 kHz. Even if signal frequencies vary within this range or contain different frequency components, the UT1000 can capture these signals. This capability enhances the sensitivity and accuracy of signal acquisition, ensuring important information or subtle changes are not overlooked. Moreover, if the device needs to extend signal acquisition to different objects emitting acoustic emission signals, a wideband sensor can adapt to these needs without requiring equipment replacement.
Regarding your concerns about modifications, you may worry that data captured by the wideband sensor could include more noise or irrelevant signal components, thus increasing the complexity of signal processing and analysis. The article addresses this by using threshold voltage to exclude noise signals; signals exceeding this threshold are identified as acoustic emission signals, while others are considered noise, effectively eliminating noise interference. Hardware design also minimizes noise as much as possible, as detailed in the next section. Overall, the design of the acoustic emission collection system, combined with the high-accuracy acoustic emission signal classification algorithm proposed in this article, ensures the equipment is not affected by unnecessary signal interference.
c) The main principle is based on a threshold. Signals with strengths exceeding the threshold voltage are identified as emission signals, while those below are considered noise. Simultaneously, efforts have been made in the hardware circuit design to minimize noise. In the power supply system modules converting 5V to 3.3V and 2.5V, the ADM7170 voltage regulator chip is selected. Its ceramic output capacitor enables rapid response to load changes and transient demands in the high-frequency range, helping the regulator maintain low noise levels in the output voltage. The OPA627 chip amplifies the original emission signal, enhancing the circuit's noise suppression capability and improving the signal-to-noise ratio of the emission signal. In wireless communication modules, Frequency-shift keying modulation also exhibits good noise resistance. Detailed explanations about these can be found on page 5, lines 164-173 of the revised manuscript.
d) The system is divided into two parts: wireless nodes and a wireless gateway. The node part consists of an acoustic emission sensor, a signal amplification circuit, an Analog-to-Digital (AD) conversion circuit, a microcontroller unit (MCU), and a LoRa wireless communication module. The MCU reads the converted acoustic emission signal data from the AD converter via a serial interface. It uses a differential recognition algorithm to determine if an acoustic emission event has occurred. Upon detecting such an event, the MCU packages the acoustic emission data and sends it to the gateway through the LoRa module.
The gateway primarily comprises a LoRa receiver and host computer software. When the LoRa receiver receives data packets from the nodes, the host computer stores the information. Once a data frame is complete, the host computer parses the detailed content of the data packet and displays it on the user interface. This process automatically extracts parameters related to acoustic emission. The figure referred to as Figure 2 in the article provides an overview of the acoustic emission acquisition system workflow.
Regarding the waveform of the acoustic emission signal, the horizontal axis represents time, and the vertical axis represents AE Data, which is the AD value of the electrical signal after ADC conversion.
e) The displayed acoustic emission waveform at this moment is obtained after the signal has undergone noise filtering by the hardware system and noise exclusion based on threshold voltage, so it does not originate from noise.
Comments 3 : [Results and discussion: How long for the testing time for each AE monitoring on different samples? What do you think that the AE data shown in this study were generated by the corrosion evolution or corrosion-induced deformation?]
Response 3: Thanks so much for your comment. Each monitoring test period corresponds to one sampling cycle. Acoustic emission data is generated by small sound signals due to corrosion, microcracks, plastic deformation, and chemical reactions inside bolts. Rusting causes changes in the metal structure on the surface and inside the bolt, such as oxidation and corrosion, thereby affecting the propagation characteristics of sound waves inside the bolt. For example, rust alters the material's speed of sound propagation and attenuation characteristics, resulting in changes to the spectrum of acoustic emission signals. Uneven roughness and irregular shapes on rusted surfaces increase the scattering and attenuation of sound waves, thereby affecting the intensity and amplitude of acoustic emission signals.
Comments 4 : [Some microscopy analysis is greatly expected to show the initial microstructure, as well as to evidence the corrosion behaviour under different corrosion levels.]
Response 4: We greatly appreciate your valuable feedback. In the revised article, we have included comparative micrographs of bolt samples showing initial microscopic rust levels at 0%, 25%, 50%, and 75%. These images were captured using a 20x magnification lens to provide a detailed comparison of rust conditions at the same positions on different samples. Different corrosion levels are primarily categorized based on electrolysis time, with specific corrosion behaviours detailed in the text. You can find corresponding content on pages 3, lines 135-136, and Figure 1.
The excerpt mentioned in the updated manuscript is “To show more vividly, Figure 1 illustrates the microscopic states of bolts with different degrees of corrosion under a magnifying glass.”

Reviewer 2 Report
Comments and Suggestions for Authors
Comments:
This paper proposes an ELM classification model based on GOOSE optimization algorithm, based on which a bolt corrosion degree classification system based on acoustic emission sensor network (WASN) was realized. Came true fast, non-destructive and accurate classification of five types of samples with 0%, 25%, 50%, 75% and 100% bolt rust degree. The classification accuracy is 98.04%, the accuracy is 98.02%, the recall rate is 98.06%, and the F1 coefficient is 0.9804.
In the method part, the innovation and implementation of GOOSE-ELM algorithm are introduced in detail, and compared with the most advanced method. In particular, the unique advantages of the proposed method in practical application are emphasized.
In the part of experimental design, the experimental design and data processing methods are described in detail, including data acquisition, experimental setting and data preprocessing steps. This will make the experimental design more transparent and repeatable.
However, there are also some problems and drawbacks, some of which are as follows.
(1) The author should carefully check the layout of the article and pay attention to the cleanliness of the article, such as formula 12 on page 12, which is out of the scope of the article frame and needs to be adjusted.
(2) The consistency of descriptive words in the figure should also be noted. For example, in the first picture, the font thickness of '75%' is not consistent with other marks, which is not beautiful enough.
(3) The HPO-ELM Predator optimization classification algorithm and SSA-ELM Sparrow optimization classification algorithm for comparison are mentioned in this paper, but only their classification results are calculated. It is suggested that the principles and advantages and disadvantages of the two algorithms should be described and evaluated simply in a concise language to make the paper more rigorous and comprehensive.
(4) The reflection and summary of the conclusion is not enough, and the classification effect is verified from the aspects of accuracy, but it does not further elaborate which aspects of the breakthrough can be further supplemented or deepened in the future, that is, the practical significance of the research needs to be elaborated.
Overall, this paper provides valuable insights into the classification of bolt corrosion states using the ELM model. Addressing the above issues can further improve the clarity, depth and impact of the manuscript, enabling it to contribute to the relevant technical field.
Comments on the Quality of English LanguageAccept after minor revision
Author Response
Reviewer #2
Firstly, we fully appreciate your careful review and valuable suggestions. We have made the following modifications in response to your suggestions:
Comments 1 : [The author should carefully check the layout of the article and pay attention to the cleanliness of the article, such as formula 12 on page 12, which is out of the scope of the article frame and needs to be adjusted.]
Response 1: First of all, thank you for pointing this out. We are very sorry for our negligence. Formula 12 on page 12 in the original manuscript has been adjusted correctly, and now it is located on page 13, which is Formula 15.
Comments 2 : [The consistency of descriptive words in the figure should also be noted. For example, in the first picture, the font thickness of '75%' is not consistent with other marks, which is not beautiful enough.]
Response 2: Thanks so much for your comment. We have modified the detail annotation you proposed and added some content to make the image description more informative and comprehensive. You can find it on page number 4, Figure 1.
Comments 3 : [The HPO-ELM Predator optimization classification algorithm and SSA-ELM Sparrow optimization classification algorithm for comparison are mentioned in this paper, but only their classification results are calculated. It is suggested that the principles, advantages and disadvantages of the two algorithms should be described and evaluated simply in a concise language to make the paper more rigorous and comprehensive.]
Response 3: Thanks so much for your comment. We have added a simple explanation of the principles, advantages and disadvantages of the HPO-ELM and SSA-ELM algorithms on page 21, lines 608-615.
The excerpt mentioned in the updated manuscript is “The HPO-ELM algorithm is inspired by carnivorous animal hunting strategies. It simulates the chase and capture process of prey to find the optimal combination in the hyperparameter space, maximizing the accuracy or other performance metrics of Extreme Learning Machine (ELM) models. It exhibits good generalization ability but may suffer from local optima. The SSA-ELM algorithm optimizes parameters in the ELM algorithm by simulating the behavior of sparrows. This enhances the algorithm's search capabilities in the solution space and its ability to discover global optima, but it has some dependence on parameter selection.”
Comments 4 : [The reflection and summary of the conclusion are not enough, and the classification effect is verified from the aspects of accuracy, but it does not further elaborate which elements of the breakthrough can be further supplemented or deepened in the future, that is, the practical significance of the research needs to be elaborated.]
Response 4: Thanks so much for your comment. We have indeed missed the reflection and summary of the conclusion in the manuscript, and we have added aspects of future complementary development in the revised manuscript on page 25, lines 694-703.
The excerpt mentioned in the updated manuscript is, “In the future, breakthroughs can be made in a comprehensive analysis of multiple parameters, integrating with other sensors for holistic assessment of bolt health, thus avoiding potential limitations of single sensors. Overall, precise monitoring of rust conditions can prevent accidents caused by bolt failures, enhancing safety and reliability. Predictive maintenance can reduce equipment downtime, further improving the availability and operational efficiency of production equipment. The application of WASN not only aids in bolt rust detection but also provides a new technological pathway for industrial automation and smart manufacturing, fostering the development of intelligent and automated production.”

Reviewer 3 Report
Comments and Suggestions for Authors
The manuscript aims to classify bolt corrosion levels using a wireless acoustic emission sensor network aided by several machine learning algorithms.
1. In this corrosion evaluation problem, I assume the active dynamic load during the testing procedure allows the sensor to capture the AE signals. What is the dynamic load?
2. The manuscript mentions five classes of bolt corrosion levels (0%, 25%, 50%, 75%, and 100%) that are manufactured and tested for classification. What is the definition of these corrosion levels? Is there a quantification equation proposed in any standard? Additionally, 12 features are used and compared, and what are their definitions?
3. The generalization of the final proposed GOOSE-ELM needs further testing. For example, can other acoustic emission signals classify the actual state of bolt corrosion? While the authors treat the problem as a classification problem, a regression problem might be more practical for representing corrosion levels, such as 60% corrosion?
4. Based on Figures 7 and 8, the clustering method seems quite effective for classifying the data. It is suggested to compare the clustering method with the proposed GOOSE-ELM.
5. Besides academic issues, there are numerous grammatical and writing errors. For instance, the equations from 13 to 26 are informal, there are grammar errors in the first paragraph of section 5.1, and Chinese characters are present in figures, etc.
6. Please check the correctness of Figures 13 and 14.
In light of these concerns, I regretfully recommend rejecting the manuscript at this time.
Comments on the Quality of English LanguageExtensive editing of English language are required.
Author Response
Reviewer #3
Firstly, we fully appreciate your careful review and valuable suggestions. We have made the following modifications in response to your suggestions:
Comments 1 : [In this corrosion evaluation problem, I assume the active dynamic load during the testing procedure allows the sensor to capture the AE signals. What is the dynamic load?]
Response 1: First of all, thank you for pointing this out. We appreciate your query and have provided further clarification in the manuscript to ensure a more precise expression. Our manuscript references the acoustic source simulation method of the stress wave tester for wood. It involves using a handheld rubber mallet to strike the upper nut, with the interaction between the rubber mallet and the upper nut serving as the sound source. Stress waves generated inside the bolt due to vibration propagate through reflection and refraction to the surface of the specimen, where AE sensors collect them. We have added specific explanations in the manuscript, which can be found on page 6, lines 191-194.
The excerpt mentioned in the updated manuscript is “At the beginning of the experiment, referring to the acoustic source simulation method of the stress wave tester for wood, use a handheld rubber mallet to strike the upper nut, and use the interaction between the rubber mallet and the upper nut as the sound source.”
Thank you for pointing out the mistake; it has been corrected.
Comments 2 : [The manuscript mentions five classes of bolt corrosion levels (0%, 25%, 50%, 75%, and 100%) that are manufactured and tested for classification. What is the definition of these corrosion levels? Is there a quantification equation proposed in any standard? Additionally, 12 features are used and compared, and what are their definitions?]
Response 2: Thanks so much for your comment. We have noted some of the novelty of this article and have cited it in our article.
In the experiment, the definition of corrosion levels is established by controlling the electrification time of bolts to categorize corrosion levels based on the resulting quality degradation of the bolts. Simultaneously, upon conclusion of the experiment, the classification results obtained are verified against the visual observation method described in the national standard, which involves assessing the coverage area and adherence status of oxidation layers on the specimen surfaces. It was found that the ratio of corroded area to the total surface area of the bolt also corresponds to the percentage defined for corrosion levels.
On page number 3, lines 109-134, a specific quantitative equation is provided to relate the theoretical mass loss of the bolts to the electrification time.
The five most basic of the 12 characteristics are explained as follows:
(1) Amplitude: Maximum voltage threshold in decibels (dB), used for wave source type identification;
(2) Rise time: Time interval between the acoustic emission signal first exceeding the threshold voltage and reaching the maximum voltage amplitude, used for noise identification;
(3) Duration: Time difference between the first and last occurrences of the acoustic emission signal exceeding the threshold voltage;
(4) Ringing count: Number of times the acoustic emission signal exceeds the threshold voltage;
(5) Power: Area under the energy envelope spectrum or the sum of squared sample values, also used for identifying the type of wave source.
It can be found on page 6, lines 208-223.
Comments 3 : [The generalization of the final proposed GOOSE-ELM needs further testing. For example, can other acoustic emission signals classify the actual state of bolt corrosion? While the authors treat the problem as a classification problem, a regression problem might be more practical for representing corrosion levels, such as 60% corrosion?]
Response 3: Thanks so much for your comment. We have carefully considered your comments, and our choice of categorization rather than regression is based on the following considerations.
According to commonly used international corrosion grade classification methods, the grades are divided into 8 levels, as follows. Level 1: Completely free from corrosion, no visible signs of rust. Level 2: Very minor rust spots do not affect the overall appearance or functionality. Level 3: Slight corrosion and slight visible rusting on the surface, but they do not affect overall performance or lifespan. Level 4: Moderate corrosion, significant visible signs of rust requiring cleaning and repair. Level 5: Severe corrosion and serious rusting affect strength and appearance. Level 6: Extreme corrosion and severe rusting requiring extensive cleaning and repair work. Level 7: Severe corrosion, partial replacement or reinforcement needed. Level 8: Extreme corrosion, total replacement and reinforcement required, significantly affecting lifespan and safety.
The degree of corrosion is typically a discrete classification problem, which better reflects the nature of practical issues. We usually classify bolts based on the severity of corrosion to determine further maintenance actions rather than attempting to predict a continuous value.
The output of a classification model is clear category labels, such as "mild corrosion" or "severe corrosion," which decision-makers find more straightforward to understand and assess. In contrast, a regression model outputs a continuous value, which requires additional interpretation and handling, making it less intuitive than categorical results.
Comments 4 : [The generalization of the final proposed GOOSE-ELM needs further testing. For example, can other acoustic emission signals classify the actual state of bolt corrosion? While the authors treat the problem as a classification problem, a regression problem might be more practical for representing corrosion levels, such as 60% corrosion?]
Response 4: Thanks so much for your comment. Your comments are very reasonable and comprehensive, so we have added the comparison between two clustering algorithms and GOOSE-ELM to the article and explained the reasons for not using the clustering algorithm according to the final results.
It can be found on page 21, lines 591-602.
The excerpt mentioned in the updated manuscript is " Based on Figures 7 and 8, the distribution of data points for different rust categories on the scatter plot can be roughly considered clustered, with noticeable gaps between clusters. Therefore, we consider using K-means clustering and hierarchical clustering methods to classify the data and compare the classification results with those of GOOSE-ELM. The classification results are shown in the table below.
Table 3. Comparison of results between two clustering algorithms and the GOOSE-ELM algorithm.
Classification method |
K-means clustering |
Hierarchical Clustering |
GOOSE-ELM |
Accuracy |
64.67 |
67.33 |
98.04 |
It can be observed that the accuracy of both clustering algorithms is not high. This is because, in practical computations, the dataset exhibits a complex structure with considerable overlap between clusters, and the gaps between different categories are not sufficiently large. Apart from the features displayed in the scatter plot, the spatial distribution differences of various rust categories are not distinct enough. Therefore, clustering algorithms are not the optimal choice."
Comments 5 : [Besides academic issues, there are numerous grammatical and writing errors. For instance, the equations from 13 to 26 are informal, there are grammar errors in the first paragraph of section 5.1, and Chinese characters are present in figures, etc.]
Response 5: Thanks so much for your comment. The equations from 13 to 23 contain formal programming terms, some of which can not be replaced by formal mathematical expressions, such as the “randi” function used in MATLAB. Other formulas have changed underscores to subscripts. Case, notation, and typography have been altered to make them more formal and concise. It can be found on pages 14-15 and 18.
We have also fixed the grammar errors and other problems in the first paragraph of section 5.1; you can find it on page 16. The excerpt mentioned in the updated manuscript is “Summarize the above process and carry out the operations. Firstly, the sample data collected by the sensor is preprocessed, normalizing it into dimensionless data. Next, utilize the ReliefF algorithm to select the most representative 7 features based on the processed data. Finally, train the optimal classification diagnostic mechanism through the GOOSE-ELM optimization and classification algorithm, ultimately achieving high-accuracy classification of input random samples.”
In addition, we have carefully considered your suggestions for further improvement, including grammar, format and professional English proofreading. To address this issue, we have invited a friend with extensive experience in academic writing to conduct a comprehensive review of the manuscript and make necessary corrections to ensure that the language used is clear, concise, and free from errors. We believe that these improvements will enhance the clarity and readability of our manuscript, making it more accessible to the intended audience. We appreciate your input and hope that these revisions meet your expectations.
Comments 6 : [Besides academic issues, there are numerous grammatical and writing errors. For instance, the equations from 13 to 26 are informal, there are grammar errors in the first paragraph of section 5.1, and Chinese characters are present in figures, etc.]
Response 6: Thanks so much for your comment. Figure13 and 14 are correct. In the initial draft, I did not clarify that 500 sample data were divided into training and testing sets in a 3:7 ratio. Now, I make it clear on page 16 of the revised manuscript, lines 490 to 493. That means the testing set comprises 150 samples. Each type of rust has 30 samples, totaling five types of rust. Therefore, in Figure 13, the x-axis represents 150 sample data points, and the y-axis represents the severity of the five types of rust. Red dots represent the actual rust category of the samples, while blue dots represent the categories assigned by the GOOSE-ELM classification. It can be seen that the three blue dots do not coincide with the red dots. This indicates one type-two rust was misclassified as type-one, one type-three rust was misclassified as type-five, and one type-four rust was misclassified as type-three.
In Figure 14, since misclassifications occurred for the second, third, and fourth types, the AUC values for these three types are not 1. Using the formula can calculate the ROC curve and further estimate the AUC based on the trapezoidal area; the final results and images are correct. The dashed line in the diagonal represents random guessing. Points on this line indicate that the classifier has no predictive capability and is just guessing randomly. Therefore, if the ROC curve is below this diagonal line, it suggests that the classifier's performance is worse than random guessing. If the ROC curve is above the diagonal line, it indicates that the classifier's performance is better than random guessing.

Reviewer 4 Report
Comments and Suggestions for Authors
This paper proposes a bolt corrosion classification system based on a Wireless Acoustic Emission Sensor Network (WASN). Focusing on achieve high prediction accuracy, an improved Goose algorithm (GOOSE) is employed to ensure the most suitable parameter combination for the ELM model. Experimental results demonstrate that the The classification accuracy obtained using the proposed method wasno less than 98.04%, significantly higher than existing methods.
However, there are still some problems in the article.
1. In Fig.5 the image is not clear, and the content needs to be written in English.
2.It is recommended to add some articles about signal denoising methods in the introduction, such as X. Lang, L. Yuan, S. Li and M. Liu, "Pipeline Multipoint Leakage Detection Method Based on KKL-MSCNN," in IEEE Sensors Journal, vol. 24, no. 7, pp. 11438-11449, 1 April1, 2024, doi: 10.1109/JSEN.2024.3364912.
Comments on the Quality of English LanguageThis paper proposes a bolt corrosion classification system based on a Wireless Acoustic Emission Sensor Network (WASN). Focusing on achieve high prediction accuracy, an improved Goose algorithm (GOOSE) is employed to ensure the most suitable parameter combination for the ELM model. Experimental results demonstrate that the The classification accuracy obtained using the proposed method wasno less than 98.04%, significantly higher than existing methods.
However, there are still some problems in the article.
1. In Fig.5 the image is not clear, and the content needs to be written in English.
2.It is recommended to add some articles about signal denoising methods in the introduction, such as X. Lang, L. Yuan, S. Li and M. Liu, "Pipeline Multipoint Leakage Detection Method Based on KKL-MSCNN," in IEEE Sensors Journal, vol. 24, no. 7, pp. 11438-11449, 1 April1, 2024, doi: 10.1109/JSEN.2024.3364912.
Author Response
Reviewer #4
Firstly, we fully appreciate your careful review and valuable suggestions. We have made the following modifications in response to your suggestions:
Comments 1 : [ In Fig.5, the image is not clear, and the content needs to be written in English.]
Response 1: First of all, thank you for pointing this out. We agree with this comment. Figure 5 has been revised by removing unnecessary information, ensuring all content is in English, and replacing it with a clearer image. This change can be found on page 7, line 227. Thank you for pointing out the mistake; it has been corrected.
Comments 2 : [ It is recommended to add some articles about signal denoising methods in the introduction, such as X. Lang, L. Yuan, S. Li and M. Liu, "Pipeline Multipoint Leakage Detection Method Based on KKL-MSCNN," in IEEE Sensors Journal, vol. 24, no. 7, pp. 11438-11449, 1 April 1, 2024, doi: 10.1109/JSEN.2024.3364912.]
Response 2: Thanks so much for your comment. We have noted some of the novelty of this article and have cited it in our article.
We have added specific steps for noise removal from the signals on page 5, line 169, referencing your publication.
The excerpt mentioned in the updated manuscript is, “It is noteworthy that in the power supply modules converting 5V to 3.3V and 2.5V, the ADM7170 voltage regulator chip is chosen to maintain low noise levels in the output voltage. The OPA627 chip is employed to amplify the original AE signals, enhancing the circuit's noise suppression capability and improving the signal-to-noise ratio of the AE signals. The frequency-shift keying modulation method is utilized in the wireless communication module, which also exhibits good noise immunity. Inspired by X. Lang et al. [16], who proposed a multiscale convolutional neural network based on kurtosis and Kullback-Leibler divergence to significantly improve noise immunity, this paper adopts a threshold voltage method to filter noise. AE signals exceeding the threshold voltage are identified as valid signals, while signals below the threshold are considered noise.”

Round 2
Reviewer 1 Report
Comments and Suggestions for Authors
First of all, I would like to apologize to all the authors and editors for keeping you waiting for the review report for so long. I was infected with a virus and rested at home for more than a week, and all my work schedules became a mess. OK, let’s go into the work. The authors paid great attention to improving the manuscript and many parts became much better, which should be appreciated. However, after carefully reading and reviewing, several points appeared to be still hard to understand and better they be improved before the manuscript becomes a paper.
Several comments were provided here. And an improved version of the manuscript is highly expected.
1. In the revised Fig. 1 with enlarged images of locals, the condition shown here should be the pre-corroded condition before AE testing, right? Can you show the image of the surface before and after AE testing? Scale bars were missed.
2. Experimental for AE measurement is not clear enough. The reviewer is still confused about how AE testing was performed in this work. Based on my understanding, the bolt samples were corroded to different levels first; then coupling the AE sensor to the bolt which was integrated with flange plate. When some dynamic excitation or collision friction was loaded, AE signals were generated and collected. Right? If so, how did the external excitation apply to the specimens? How long was the excitation? What does it mean “one sampling cycle” for the AE test period.
3. Corrosion was considered as one possible AE source. But this is too general. Corrosion is a complex process including various physical and chemical changes. Eg., the surface passive film breakage, deposition of corrosion products or rust, the cracking behavior within the rust, hydrogen-bubble evolution, etc. The author should discuss it more specifically.
4. Plastic deformation: what external excitation caused the plastic deformation? If so, it is easy to find evidence images by OM or SEM.
5. Some critical information seems missing yet. What are the threshold voltage setting and preamplifier gain for AE testing? How to define or individualize an AE waveform in the AE acquisition system? Some parameters, such as the hit definition/lockout time, should be included.
6. Regarding the choice of AE sensor, it is not wise at all to refer to some reference reports. By referring to the previous report, the exact value range of AE frequency in relation to the bolt corrosion or deformation was very unreliable. Actually, this is very easy to identify. Performing several tests with a wide band sensor for specimen in blank test, working condition, and corrosion condition, or deformation condition, one can easily get the general range of the relevant values. On the other hand, it is not a worry on the detection of too wide-frequency-range AE signals including noises using wideband sensor. Instead, the sensitivity is the critical point. From the discussion in the manuscript, it shows the main AE frequency covering 0-150 kHz. Given this is true, it is so narrow that the first choice of AE sensor should be a resonant AE sensor like R15a (Mistras group, 150 kHz resonant frequency). Because a wide band sensor usually has a lower sensitivity than the resonant sensor. But the sensitivity of AE sensor is the Achille' Heel for AE testing.
7. It is critical to ensure the validity of the AE data. If the data itself is flawed, such as from noise, then all the fancy analysis using AI or machine learning that follows is nonsense.
Author Response
Reviewer #1
Firstly, I was truly sorry to hear that you weren't feeling well recently, but I'm relieved to know that you're on the mend now. Your health and well-being are of utmost importance, and I'm glad to hear that you're recovering. We appreciate your hard work during the recovery period and your valuable comments on our manuscript. We have made the following modifications in response to your suggestions:
Comments 1 : [ In the revised Fig. 1 with enlarged images of locals, the condition shown here should be the pre-corroded condition before AE testing, right? Can you show the image of the surface before and after AE testing? Scale bars were missed.]
Response 1: First of all, thank you for pointing this out. I may not have understood exactly what your comment meant, but I will try my best to explain and elaborate on them to you, if I have not reached the designated position also please more inclusion. Fig. 1 with enlarged images of locals describes the enlarged local map corresponding to the bolt samples with five kinds of corrosion degrees prepared before AE testing. Therefore, it is not the pre-corroded condition, but already the degree of corrosion is 0%/25%/50%/75%/100% respectively. ‘0%’corresponds to the initial state that the bolt has not been energized and has no rust on the surface. After AE testing, the degree of corrosion of bolts will not change, so we can't show different contrast graphs of surface status before and after AE testing. To reply to you in more detail and rigorously, the acoustic emission waveforms of 25%, 50%, 75%, and 100% rust are shown below for you to further understand the experimental process.
Figure 1. Schematic diagram of AE waveforms. (a) AE waveforms of bolts with corrosion levels of 25%; (b) AE waveforms of bolts with corrosion levels of 50%; (c) AE waveforms of bolts with corrosion levels of 75%; (d) AE waveforms of bolts with corrosion levels of 100%.
The results show that the sample data of the high-rusted bolt are more dense and the AE feature points are more. Taking the acoustic emission waveform of 75% rust in Figure c as an example, there are two places in the waveform where the feature points are obviously gathered in large numbers, that is, the bolt sample with 75% rust degree at this time has two places where the rust is particularly serious.
Regarding Scale bars, I added clarity about the magnification of the magnifying glass on page 3, line 139.
Comments 2 : [Experimental for AE measurement is not clear enough. The reviewer is still confused about how AE testing was performed in this work. Based on my understanding, the bolt samples were corroded to different levels first; then coupling the AE sensor to the bolt which was integrated with flange plate. When some dynamic excitation or collision friction was loaded, AE signals were generated and collected. Right? If so, how did the external excitation apply to the specimens? How long was the excitation? What does it mean “one sampling cycle” for the AE test period.]
Response 2: Thanks so much for your comment.
Your understanding of the AE testing is correct. I would like to respond further to your questions. On page 6, lines 201 to 206, it is explained how the external excitation acts on the specimen. Based on the sound source simulation method widely used in wood stress wave testers, the upper nut is struck with a hand-held rubber mallet, and the interaction between the rubber mallet and the upper nut is used as the sound source. In addition, the difference recognition algorithm was also used in the design of WASN. The difference between the amplitude of the end time and the amplitude of the initial time is calculated by using the data of the acoustic emission signal at a fixed time to determine whether the threshold is exceeded. In this way, we can further judge whether acoustic emission events occur at present. Therefore, the difference in characteristics such as the duration caused by the strength of the external excitation (the size of the hammering force) does not affect the final classification result. At the same time, the algorithm can effectively filter out the interference of external sharp noise.
Figure 4 has been added to the latest manuscript to show the reader more intuitively how external incentives act on the specimen.
The sampling period is 1s. 10,000 samples are taken in this second. Therefore, the horizontal coordinate ranges from 0 to 10000, as shown in Figure 1 (d). The reason why 10,000 is not shown in (a), (b), and (c) in Figure 1 is that the effective acoustic emission signal duration is short and does not reach the 1s clock. That is, the collected acoustic emission signals exceeding the threshold value do not exceed 10,000. This part of the content has also been added to the revised version of page 7, lines 246-249.
Comments 3 : [Corrosion was considered as one possible AE source. But this is too general. Corrosion is a complex process including various physical and chemical changes. Eg., the surface passive film breakage, deposition of corrosion products or rust, the cracking behavior within the rust, hydrogen-bubble evolution, etc. The author should discuss it more specifically.]
Response 3: Thanks so much for your comment. Your suggestions are very comprehensive and rigorous, and are worth thinking and discussing. However, this paper is mainly to identify and classify the overall corrosion state of the current bolt, to facilitate the decision maker to make the next step, rather than analyzing the corrosion principle in detail. The discussion on the specific causes of rust and the evolution process is a good inspiration for us, and we will carry out further research. Thank you for your suggestions. But this article is only for accurate classification.
Comments 4 : [Plastic deformation: what external excitation caused the plastic deformation? If so, it is easy to find evidence images by OM or SEM.]
Response 4: Thanks so much for your comment. Based on our review of the information, corrosion of bolts usually does not fall into the category of plastic deformation. Plastic deformation refers to the permanent shape or size change of the material during the stress process, which occurs at the atomic and molecular level inside the material and is caused by external forces exceeding the yield point of the material, involving the rearrangement of the internal structure of a material. Bolt rust refers more to the surface change or damage caused by chemical reactions or corrosion because of exposure to wet and corrosive substances in the environment, usually does not involve the plastic deformation of the internal structure of the material, but makes the surface rust or corrosion, further leading to the strength and durability of the bolt. Therefore, it is not necessary to use OM or SEM to explore such microscopic structures.
Comments 5 : [Some critical information seems missing yet. What are the threshold voltage setting and preamplifier gain for AE testing? How to define or individualize an AE waveform in the AE acquisition system? Some parameters, such as the hit definition/lockout time, should be included.]
Response 5: Thanks so much for your comment. In response to your questions, I will respond to each one. The selection of the threshold in this paper is determined by using the statistical characteristics of signal and noise in the process of AE testing. The threshold value is set according to the statistical parameters such as mean value and variance of signal and the distribution of noise. According to the characteristics of different specimens, the performance under different thresholds was observed, and the optimal threshold was found by repeated adjustment. An explanation of the threshold selection is added on page 7, lines 224-226.
The circuit design of the OPA627 facilitates early test adjustment of the required sound amplification. In the test, the gain of the amplifier circuit is first adjusted 10 times, and then the function signal generator is used to output the sine wave signal with the same amplitude and different frequency, and then the signal is connected to the input end of the amplifier circuit to test the amplifier circuit performance. The output waveform at the output end of the amplifier circuit is measured by an oscilloscope, and the output amplitude of the output waveform is observed to determine whether the corresponding amplification times are reached. In the actual test, the voltage peak-to-peak value (VPP) of the input sine wave signal is set to 100mV, and then the frequency of the sine wave is set to 1KHZ, 100KHZ, 500KHZ, 1MHZ, 2MHZ, and the measurement waveform of the oscilloscope is observed. The test results of selecting 1MHZ and 2MHZ are shown in Figure 2.
Figure 2. Effect of signal amplification at different frequencies (a) Output waveform of 1MHZ signal (VPP=1V); (b) Output waveform of 2MHZ signal (VPP=781.3mV).
By observing the peak-to-peak value of the output waveform through the waveform diagram, it can be seen that when the input frequency is less than 1MHZ, the amplifier circuit can effectively amplify the input signal, and the amplification factor is 10 times. Only when the output signal exceeds 1MHZ, the output signal will be attenuated. Therefore, in the frequency range of the bolt acoustic emission signal, both the bias current voltage and the input impedance, the amplifier performance of the OPA627 is excellent. However, since the specific discussion of the hardware part is not the focus of this article, we do not choose to add the presentation of this part in the revised draft, I hope you can understand.
The definition and eigenvalue extraction of acoustic emission waveform in the acoustic emission acquisition system is to first find the maximum and minimum values in the acoustic emission signal data, record the maximum position, and calculate the signal amplitude. Secondly, the AE signal power is calculated, that is, the area between the waveform and the Y-axis. Thirdly, the first and last exceeded threshold locations are searched, and the duration and rise time of the acoustic emission signal are calculated. Finally, the ring location is searched in the interval and the total number of rings is recorded. In this process, the location information of the key points of the waveform is saved and marked on the waveform diagram of the acoustic emission signal. Finally, to facilitate data classification and damage identification later, the AE signal feature value information is stored in Excel in a fixed format. All the definitions of the parameters used are set out in the latest manuscript on page 7, lines 224-237.
Comments 6 : [Regarding the choice of AE sensor, it is not wise at all to refer to some reference reports. By referring to the previous report, the exact value range of AE frequency in relation to the bolt corrosion or deformation was very unreliable. Actually, this is very easy to identify. Performing several tests with a wide band sensor for specimen in blank test, working condition, and corrosion condition, or deformation condition, one can easily get the general range of the relevant values. On the other hand, it is not a worry on the detection of too wide-frequency-range AE signals including noises using wideband sensor. Instead, the sensitivity is the critical point. From the discussion in the manuscript, it shows the main AE frequency covering 0-150 kHz. Given this is true, it is so narrow that the first choice of AE sensor should be a resonant AE sensor like R15a (Mistras group, 150 kHz resonant frequency). Because a wide band sensor usually has a lower sensitivity than the resonant sensor. But the sensitivity of AE sensor is the Achille' Heel for AE testing.]
Response 6: Thanks so much for your comment. For the frequency band range of bolt acoustic emission signal, our measured results are in agreement with reference reports. To show stronger basis, reference reports are cited in the paper. Your suggestion on the selection of sensors for acoustic emission equipment is very correct and rigorous. we very much agree with it. We also have acoustic emission devices in our lab that use R15 sensors. Only the previous experiments found that there was no big difference in the results between the two, so no deliberate distinction was made. Our laboratory used the UT1000 acoustic emission device to detect the acoustic emission signal of wood in the early stage, so it is directly used in this article. In the future, we will carefully consider your suggestions and carefully distinguish between them.
Figure 3. Wireless acoustic emission device based on R15 sensor.
Comments 7 : [It is critical to ensure the validity of the AE data. If the data itself is flawed, such as from noise, then all the fancy analysis using AI or machine learning that follows is nonsense.]
Response 7: Thank you for your comments. We admire your rigorous academic attitude. Please rest assured that our acoustic emission devices have been studied and used in many aspects such as rail and wood, and the results are accurate and feasible, and relevant papers have been successfully published. Abnormal data are screened and processed during the experiment, so you don't have to worry too much about its validity.
We sincerely hope that the revised version meets your expectations and satisfies the requirements for publication. We believe that the changes we have made significantly improve the quality of the paper and address the concerns raised in your review.
We would be grateful for your consideration of the revised manuscript. We are eager for the opportunity to publish this work and contribute to the field.
Thank you once again for your time and effort.

Reviewer 3 Report
Comments and Suggestions for Authors
1. In Table 3 of the revised manuscript, what is the definition of accuracy? Is the definition consistent across all methods?
2. "The degree of corrosion of bolts is divided and prepared according to the different durations of energizing. The bolt corrosion degree is 100% when the power is on for 30 hours, and the bolt corrosion degree is 0%, 25%, 50%, and 75% when the power is on for 0, 6, 12, 18, and 24 hours, respectively." If a standard exists for these measurements, it should be cited. Additionally, the response regarding corrosion levels mentions area ratio and mass loss, while the revised manuscript references immersion time, which is confusing. Please clarify.
3. There are Chinese characters in Figure 15.
4. The ROC curve in Figure 14 is linear. Please provide an explanation for this.
Comments on the Quality of English LanguageModerate editing of English language required.
Author Response
Reviewer #3
Firstly, Thank you very much for your insightful comments and valuable suggestions on our manuscript. We have carefully addressed all the points raised and have revised the manuscript accordingly.
Comments 1 : [In Table 3 of the revised manuscript, what is the definition of accuracy? Is the definition consistent across all methods?]
Response 1: Thanks so much for your comment for pointing this out. The definition of accuracy, which is used to evaluate all methods is consistent, and accuracy is the ratio of the number of samples correctly predicted to the total number of samples, which is consistent throughout. In Table 3, the accuracy of the clustering method is obviously low because the clustering method is not very suitable for the classification problems in this paper, which leads to a large difference in accuracy.
At the same time, we did not give a clear definition of accuracy in the article, because we regard it as a common sense problem that is easier to understand. Now we have given an accurate definition and formula in lines 522 and 531-532 on page 18 of the latest revised draft. Thank you for your suggestion.
Comments 2 : ["The degree of corrosion of bolts is divided and prepared according to the different durations of energizing. The bolt corrosion degree is 100% when the power is on for 30 hours, and the bolt corrosion degree is 0%, 25%, 50%, and 75% when the power is on for 0, 6, 12, 18, and 24 hours, respectively." If a standard exists for these measurements, it should be cited. Additionally, the response regarding corrosion levels mentions area ratio and mass loss, while the revised manuscript references immersion time, which is confusing. Please clarify.]
Response 2: Thanks so much for your comment. As for how the corrosion grade is divided in the process of specimen preparation, we make the following explanation. "Energized accelerated rust experiment" is the rule that we follow in the preparation process. Its core is to distinguish the degree of rust through the difference in energized time. In this process, the specimen will inevitably produce quality loss, which is proportional to the energized time, and is a parameter involved in the experiment content, which does not contradict the central idea of the experiment. As for the area ratio mentioned, this is the international standard judgment method of bolt rust degree, which is used to verify the accuracy of the classification results of the "electric-accelerated rust experiment" in the article.
This may not have been explained clearly in the previous manuscript, but it has now been further explained in the latest manuscript on page 3, lines 133-137.
Comments 3 : [There are Chinese characters in Figure 15.]
Response 3: Thank you for your reminder. I'm sorry for the error. We found this problem when we submitted the first version of the manuscript and corrected it, but there may have been a mistake at the time of submission and we did not update the correct version. Now we have corrected it twice.
Comments 4 : [The ROC curve in Figure 14 is linear. Please provide an explanation for this.]
Response 4: Thanks so much for your comment. The reason the ROC curve appears as a polyline at both ends is that the classification results of the model (0 or 1) are used directly when the ROC curve is drawn, rather than the probability values. When the model outputs a classification result of 0 or 1, these results are directly used to calculate the false positive rate and the true positive rate, thus generating points on the ROC curve. Because these points are discrete and often reached through different thresholds, the curve formed by connecting these points appears graphically as a polyline.
There are certain advantages to using a polyline ROC curve, especially when the discrete classification results need to be evaluated directly. The polyline ROC curve directly reflects the classification performance of these discrete results and can intuitively show the performance of the model under a specific threshold. It helps clarify the effect of the division between different categories. The comparison of each category to the others clearly shows the difference in the performance of the classifier, particularly in cases where the model may have significantly improved on only a few key categories. Polyline ROC curves simplify calculations, provide direct decision feedback, and better reflect classification performance for specific business needs.
Thank you for your valuable feedback on our manuscript, particularly regarding the English writing quality. Our team has put considerable effort into optimizing and improving the language, enlists the help of a veteran overseas professor to improve clarity and readability.
We hope that these revisions meet your expectations and improve the overall quality of the manuscript. We are eager for the opportunity to publish this work and contribute to the field.
Thank you once again for your time and effort.

Round 3
Reviewer 1 Report
Comments and Suggestions for Authors
I would like to thank the authors for their detailed responses to my comments. My questions and concerns have been well answered and addressed. Based on the careful reading and review of the full manuscript, I cautiously and firmly recommend it for publication in its present form.
Also, I encourage the authors to continue the relevant studies and look forward to better results and findings.